# RNA targeting and cleavage by the type III-Dv CRISPR effector complex

Evan A. Schwartz [1,10], Jack P. K. Bravo [2,10], Mohd Ahsan [3,10], Luis A. Macias [4], Caitlyn L. McCafferty [1], Tyler L. Dangerfield [2], Jada N. Walker [4], Jennifer S. Brodbelt [4], Giulia Palermo [3]✉, Peter C. Fineran [5,6,7], Robert D. Fagerlund [5,6,7]✉ & David W. Taylor [1,2,8,9]✉

CRISPR-Cas are adaptive immune systems in bacteria and archaea that utilize CRISPR RNA-guided surveillance complexes to target complementary RNA or DNA for destruction[1–5]. Target RNA cleavage at regular intervals is characteristic of type III effector complexes[6–8]. Here, we determine the structures of the *Synechocystis* type III-Dv complex, an apparent evolutionary intermediate from multi-protein to single-protein type III effectors[9,10], in pre- and post-cleavage states. The structures show how multi-subunit fusion proteins in the effector are tethered together in an unusual arrangement to assemble into an active and programmable RNA endonuclease and how the effector utilizes a distinct mechanism for target RNA seeding from other type III effectors. Using structural, biochemical, and quantum/classical molecular dynamics simulation, we study the structure and dynamics of the three catalytic sites, where a 2′-OH of the ribose on the target RNA acts as a nucleophile for in line self-cleavage of the upstream scissile phosphate. Strikingly, the arrangement at the catalytic residues of most type III complexes resembles the active site of ribozymes, including the hammerhead, pistol, and Varkud satellite ribozymes. Our work provides detailed molecular insight into the mechanisms of RNA targeting and cleavage by an important intermediate in the evolution of type III effector complexes.

Many bacteria employ type III clustered regularly interspaced short palindromic repeats (CRISPR)-Cas (CRISPR-associated) systems as an adaptive immune response to bacteriophage infection[1–3]. Upon binding an invasive RNA transcript, type III CRISPR-Cas complexes induce cyclic oligoadenylate (cOA) production by the palm domain of Cas10[11–14]. Cyclic oligoadenylates are allosteric activators of ancillary factors including proteases and the Csm6 and NucC nucleases, which

bolster the immune response[11,12,15,16]. Furthermore, binding of a non-self RNA target initiates ssDNA cleavage using the HD domain of Cas10[17–19]. After hybridization to the crRNA, type III CRISPR-Cas complexes also cleave bound RNA transcripts with unique periodicity by multiple Cas7 subunits (every 6 nucleotides for III-A, III-B, and III-E systems)[4,6,8,20]. Cleavage and release of a non-self RNA target arrests cOA production and ssDNA cleavage. The RNA-dependent activation of accessory

[1]Interdisciplinary Life Sciences Graduate Programs, University of Texas at Austin, Austin, TX, USA. [2]Department of Molecular Biosciences, University of Texas at Austin, Austin, TX, USA. [3]Department of Bioengineering and Department of Chemistry, University of California, Riverside, CA, USA. [4]Department of Chemistry, University of Texas at Austin, Austin, TX, USA. [5]Microbiology and Immunology, University of Otago, PO Box 56 Dunedin, New Zealand. [6]Bioprotection Aotearoa, University of Otago, PO Box 56 Dunedin, New Zealand. [7]Genetics Otago, University of Otago, PO Box 56 Dunedin, New Zealand. [8]Center for Systems and Synthetic Biology, University of Texas at Austin, Austin, TX, USA. [9]LIVESTRONG Cancer Institutes, Dell Medical School, University of Texas at Austin, Austin, TX, USA. [10]These authors contributed equally: Evan A. Schwartz, Jack P. K. Bravo, Mohd Ahsan. ✉e-mail: gpalermo@engr.ucr.edu; robert.fagerlund@otago.ac.nz; dtaylor@utexas.edu

nucleases by cOAs has recently been exploited for a range of sequence-specific diagnostic tools, including SARS-CoV-2 detection[21–23].

Type III CRISPR-Cas complexes have been proposed as ancient ancestors of CRISPR-Cas systems, with Cas10 predicted as the first CRISPR-associated protein[24]. These systems are also widespread, particularly in archaea, and diverse in terms of their gene composition and organization. This evolutionary diversity has recently been highlighted by the type III-E system, which contains an operon that features a single polypeptide effector consisting of multiple Cas7 subunit domain fusions, including one domain split by a large insertion[9,10]. However, the type III-E system lacks the Cas10 and Cas5 subunits characteristic of cOA production and crRNA binding in other type III systems, respectively. While the evolutionary pathway between the well-characterized multi-subunit type III effector complexes and the III-E effector is not entirely understood, the type III-D systems may represent an evolutionary intermediate[9,10]. The type III-D systems are marked by the presence of *csx10* (a specific variant of *cas5*) and often have a *csx19* gene of unknown function. One previous report highlighted the evolutionary progression from multi-gene systems (III-D1) to the single-subunit type III-E system[9]. Recently, a variant type III-D system (III-Dv) was annotated with multiple gene fusions, suggesting its role as an evolutionary intermediate between the multi- and single-subunit type III effectors (Supplementary Fig. 1a)[10]. Type III-Dv also includes the large insertion interrupting the terminal *cas7* gene observed in type III-E system[9,10,25].

The mechanism of RNA target cleavage by type III effector complexes has been well studied. It was initially demonstrated that cleavage products have a 5′-OH and 2′,3′-cyclic phosphate, suggesting that cleavage was divalent metal ion-dependent[4,8]. Subsequent studies demonstrated that mutation of the putative ion-coordinating Cas7 aspartate residue or removal of the target RNA 2′-OH abrogated cleavage, indicating that this cleavage mechanism was RNA catalyzed but required Cas7 as a protein factor to promote hydrolysis of the RNA target phosphodiester bond[26–28]. However, the essential catalytic $Mg^{2+}$ ion had not been directly observed in any structures of a type III effector in complex with target RNA, leaving ambiguity about the mechanism of catalysis. Here, we present cryo-electron microscopy (cryo-EM) structures of the type III-Dv complex in four distinct states: bound to a crRNA (binary, surveillance complex), bound to an RNA non-self target in a pre-cleavage state, and bound to a self-target with coordinated magnesium in pre-cleavage and post-cleavage states. Through analysis of structural rearrangements between the binary and RNA target-bound complex, we show how structural rearrangements create a strict seed requirement for RNA binding and activation. We demonstrate programmable RNA cleavage at three separate active sites across three unique Cas7 subunits. Careful examination of the active sites of our structure uncovers conserved acid-base catalysis that facilitates in-line scissile phosphate self-cleavage by the 2′-OH group positioned one nucleotide downstream on the RNA target, displaying remarkable similarities to known ribozymes.

## Results

### The type III-Dv effector forms a 332 kDa complex with no repeated subunits

The operon of the type III-Dv complex from *Synechocystis* sp. PCC6803 contains *cas10*, a *cas7-cas5-cas11* fusion, a double *cas7* fusion (*cas7-2x*), *csx19*, and an insertion-containing *cas7* (*cas7-insertion*). Downstream of the *cas* operon is *cas6-2a*, adaptation genes, and a CRISPR array containing 56 spacers ranging in size of 34 to 46 bp, and upstream are genes encoding accessory proteins Csm6 and Csx1[29,30]. To determine the composition of the type III-Dv complex, we cloned the *cas* operon, *cas6-2a*, and first repeat-spacer-repeat of the CRISPR array from *Synechocystis* and expressed them in *E. coli* (Fig. 1). The complex was purified using metal affinity and size-exclusion chromatography, where it eluted at ~330 kDa (Supplementary Fig. 2). Analysis of the purified complex by

SDS-PAGE and mass spectrometry confirmed the presence of all proteins except for Cas6-2a, consistent with other multi-subunit type III complexes[6,7] (Fig. 1c, d, Supplementary Table 1). The presence of Csx19 indicates this protein is a core component of the type III-Dv complex. Urea-PAGE analysis showed a single mature crRNA with a length of 37-nt (Fig. 1b), as previously reported for type III-Dv crRNAs extracted from *Synechocystis*[29]. Native mass-spectrometry revealed that a single copy of each subunit assembled into a 332 kDa complex (Fig. 1d, Supplementary Table 1), in contrast to type III-A and -B complexes which contain multiple copies of Cas7 and Cas11 subunits[13,14,26,28,31,32]. The equal stoichiometry of subunits in the type III-Dv complex may explain why they assemble around a fixed length crRNA, while other type III complexes can assemble around crRNAs of varying lengths[6].

### Structures of the type III-Dv surveillance complex

We determined the cryo-EM structure of the type III-Dv surveillance complex containing the 37-nt crRNA at a global resolution of 2.5 Å, enabling us to fit and refine AlphaFold2-predicted models and rapidly generate a complete atomic model for the complex (Fig. 1e–g, Supplementary Table 2 and Supplementary Figs. 3, 4, 5). The overall core architecture largely resembles the recently determined type III-E[33,34] surveillance complexes, with the insertion domain protruding from the top of the complex (Supplementary Fig. 1b). A notable exception is the presence of the Cas10 and Csx19 subunits at the base of the type III-Dv complex. An uncharacterized domain comprising the N-terminal 112 residues of the Cas7-insertion subunit was absent from our map, likely due to flexibility. Despite the different subunit stoichiometries compared with other type III complexes, our structure revealed that the fused subunits still arrange into a repeating backbone around the crRNA, a structural feature conserved across all class 1 CRISPR-Cas complexes. Furthermore, all Cas7, Cas5, Cas11, and Cas10 domains within the type III-Dv complex align very well with their type III-A and III-B counterparts (Supplementary Fig. 5i–k). Running along the major Cas7 filament is the Cas11 domain of Cas7-Cas5-Cas11 and the C-terminus of Cas10, both of which are highly alpha-helical and resemble canonical small subunits[35,36]. Csx19, a subunit unique to type III-D1 and III-Dv, contains multiple β-sheets and is nestled at the base of the effector between Cas5 and Cas10 subunits, making contacts with the crRNA, and contributing to the structural integrity of the complex (Supplementary Fig. 5f, g). Indeed, affinity purification of a ΔCsx19 complex with an N-terminal Cas10 tag did not result in pulldown of the complex, indicating that assembly of Csx19 onto the type III-Dv complex is essential for complex assembly and stability (Supplementary Fig. 2b, d).

It has been hypothesized that the type III-Dv complex represents an evolutionary intermediate between the multi-subunit type III-D1 system and the recently described single polypeptide type III-E system (Supplementary Fig. 1)[10]. Our structure supports this evolutionary relationship, as the organization of the type III-D1 operon is maintained and subunits are physically fused together through flexible ~20-residue linker polypeptides. This is exemplified by the Cas7-Cas5-Cas11 subunit, which adopts a tortuous topology that places Cas7 in the body of the complex, Cas5 below it, and Cas11 back on top of Cas7 (Fig. 1g). While this demands long linkers connecting the domains, we hypothesize that this enables the modular folding of each Cas domain to ensure accurate complex assembly and did not require genetic shuffling of the gene order in the type III-Dv operon. Similarly, type III-E also possesses long linkers between Cas7 and Cas11 domains. Overall, our structure suggests that the type III-Dv complex represents an intermediate, both genetically and structurally, between multi-subunit and single-subunit type III systems.

### RNA targeting by the type III-Dv complex

To understand the molecular mechanism of RNA targeting, we determined the cryo-EM structure of the type III-Dv effector in complex with

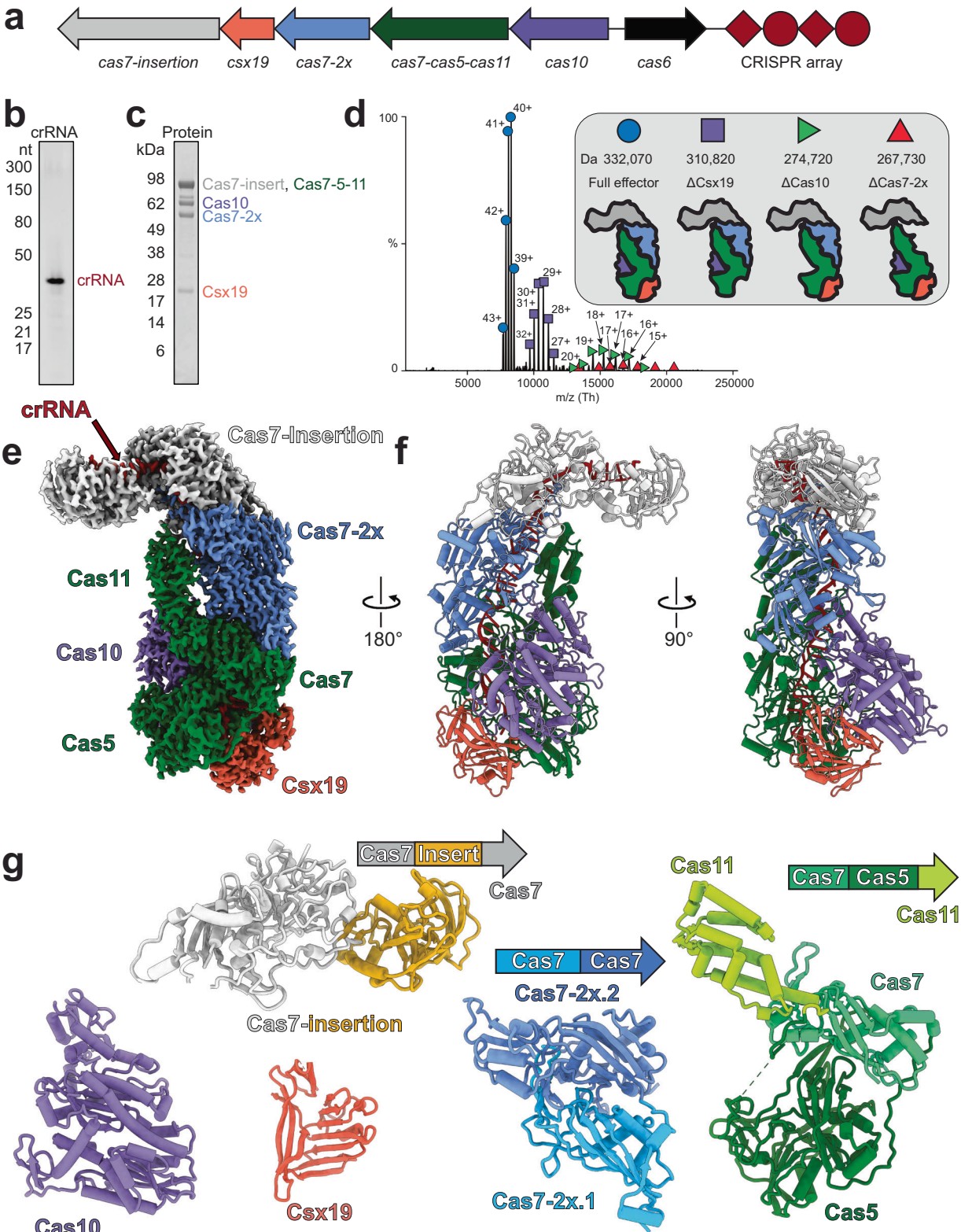

**Fig. 1 | Stoichiometry and architecture of the *Synechocystis* CRISPR-Cas type III-Dv effector. a** Gene organization of the type III-Dv operon. Subunits and nucleic acids are colored as follows: *cas10*, purple; *cas7-cas5-cas11*, green; *cas7-2x*, blue; *cas7-insertion*, white. Cas6 is not found in the structure or purification and is thus colored black. Adaptation and accessory enzyme genes have been removed for simplicity. **b** TBE-Urea PAGE analysis of crRNA length for the type III-Dv complex. The crRNA product is 37-nt long. Representative of two independent experiments. **c** SDS-PAGE of the purified type III-Dv complex after size-exclusion chromatography. Representative of two independent experiments. **d** Native MS-MS of the type III-Dv complex. Peaks correspond to the full WT complex (circle), ΔCsx19 (square), ΔCas10 (green triangle), and ΔCas7-2x (red triangle). cCharge states are labeled for each peak. **e** Cryo-EM map of type III-Dv binary complex. Subunits are colored as in the operon. **f** Atomic model of the type III-Dv binary complex. **g** Models of each subunit in the type III-Dv complex, colored by domains. Source data are provided as a Source data file.

a 60 nucleotide (nt) target RNA at 2.8 Å resolution (Fig. 2a). The structure appears nearly identical to the binary complex, except for the presence of 34-nt of the RNA target hybridized to the crRNA. As in other class 1 complexes, every 6th nucleotide of the crRNA and RNA target is flipped out by the β-hairpin thumb domain of each Cas7 domain, with the exception for the Cas7-insertion subunit[26,28,33,35,37–42]. Instead, the Cas7-insertion subunit threads an unstructured loop, 4-nt upstream of the previous Cas7, between bases U29 and C30 of the crRNA and G27 and A28 of the RNA target, pushing the stacking bases apart (Supplementary Fig. 5l). The crRNA and RNA target at this position does not have the same kinked geometry as the other Cas7 sites in

the complex, suggesting that this position of the RNA target may not be designated for cleavage.

In our structure, while the target RNA engages in Watson-Crick base pairing along almost the entirety of the crRNA, after position C8 in the crRNA, the target RNA disengages at the anti-repeat sequence and is funneled into an exit channel on the surface of Cas10 (Supplementary Fig. 6a). This is reminiscent of non-self RNA target recognition that occurs within the Cas10 subunits of type III-A and III-B effector complexes[13,14,28]. Comparison of the target-bound complex with the binary, surveillance complex shows only minor conformational changes throughout the Cas7 backbone. However, there are notable

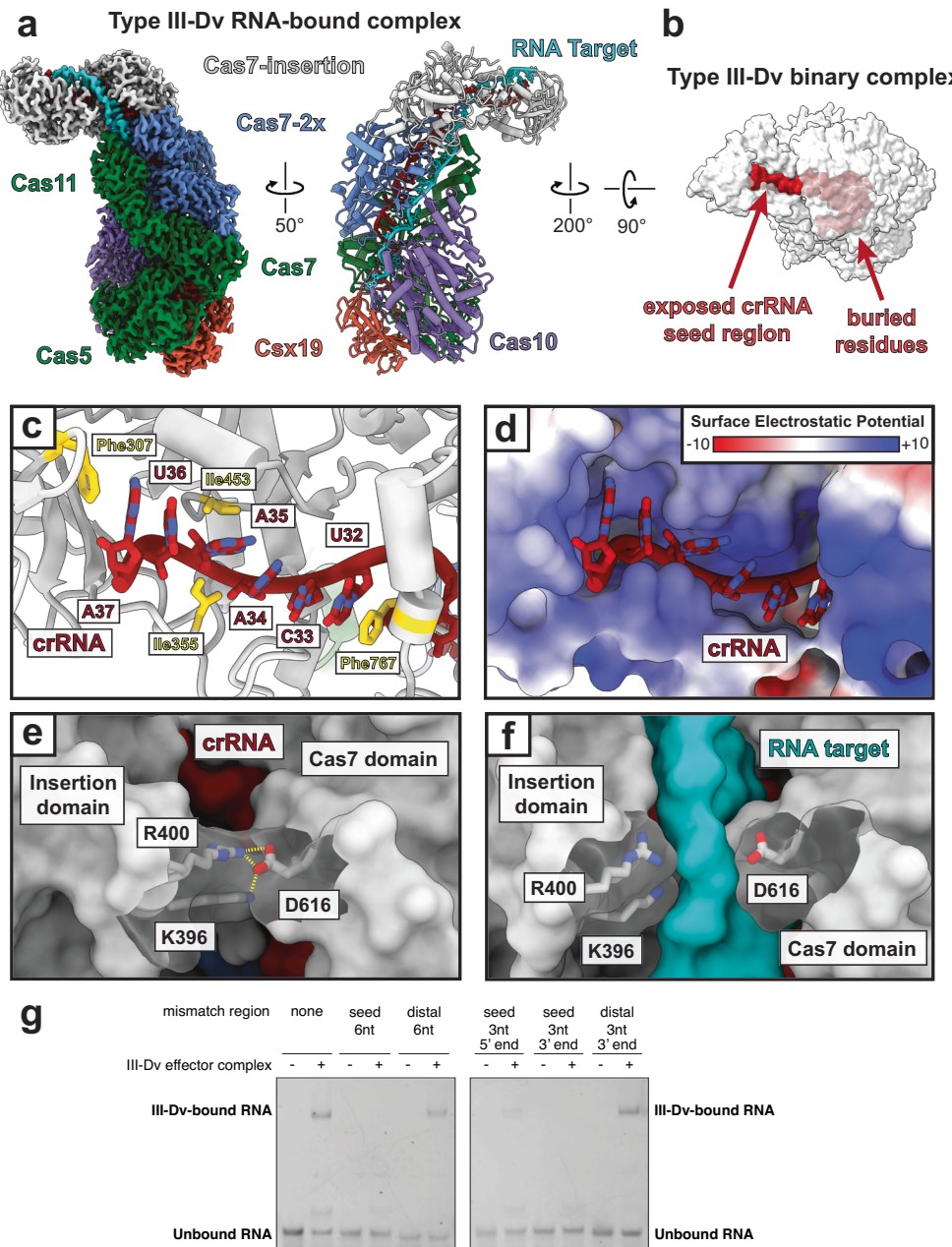

**Fig. 2 | Exposed crRNA seed region initiates RNA target binding. a** Structure of the type III-Dv (ternary) complex bound to a target RNA, with the cryo-EM map on the left and atomic model on the right. **b** Surface representation of the type III-Dv complex highlights the buried surface of the crRNA except for a seed region that is exposed by the insertion domain of Cas7-insertion. **c** Exposed residues in the crRNA seed region stabilized by residues in the insertion domain. **d** The crRNA seed region sits in a positively charged pocket of the insertion domain. Units are in kT/e. **e** A salt bridge between D616 and R400/K396 blocks RNA target binding at this region, presumably requiring seeding first. Dashed lines show the salt bridge interactions. **f** Separation of the salt bridge in e to accommodate the RNA target.
**g** Electrophoretic Mobility Shift Assay (EMSA) to test binding of RNA targets containing mismatches with the crRNA in the target 5′ seed region and 3′ end. Representative of two independent experiments. Source data are provided as a Source data file.

rearrangements in the Cas10 subunit. Closer inspection of the two structures reveals that an alpha helix (L238-F245) must be displaced to accommodate the 3′ end of the target RNA strand through the exit channel within Cas10, which is a potential feature of Cas10 activation (Supplementary Fig. 6a, b).

While most of the crRNA spacer is completely buried within the type III-Dv complex, six bases at the 3′ end are highly exposed in the insertion domain of the Cas7-insertion subunit (Fig. 2b). The positive surface of the "Insertion" domain forms a pocket where the crRNA backbone is cradled, while the crRNA bases U(32) and A(37) are held in place through stacking interactions with F767 and F307 of Cas7-insertion, respectively (Fig. 2b–d, Supplementary Fig. 5d, e). The crRNA 3′ structure is further stabilized through hydrophobic contacts with I355 and I453. These interactions capture the 3′ end of the crRNA in a rigid conformation where the Watson-Crick faces are solvent accessible, and thus amenable to efficient base pairing. This mechanism likely shares similarities to the seeding mechanism of the type III-B complex, where the 3′ end of the crRNA must hybridize with the target RNA before the rest of the crRNA is accessible for hybridization[22]. An electrostatic network between R400, K396, and D616 within the Cas7-insertion subunit joins the cleft between the insertion domain and the Cas7 domain, limiting RNA hybridization with the crRNA (Fig. 2e). This salt-bridge is ruptured upon faithful RNA target annealing (Fig. 2f).

To test whether the exposed 3′ region of the crRNA acts as a seed for RNA targeting, we determined RNA binding to RNA targets containing various mismatches in this region through electrophoretic mobility shift assays (EMSAs). The presence of 3- and 6-nt mismatches between the RNA target and the 3′ end of the crRNA within the solvent-exposed region of the insertion domain largely prevented binding, with the seed region closest to the cleft being the most sensitive to mismatches (Fig. 2g). Conversely, mismatches at the distal, 3′ end of the RNA target had little effect on binding. These results implicate the solvent exposed region of the crRNA as the seed critical for propagation of the crRNA:target sequence duplex.

This data suggests that the type III-Dv complex utilizes a mechanism for target seeding distinct from previously-characterized type III complexes. Type III-A and -B complexes have variable mature 3′ crRNA ends due to the processing by cellular housekeeping nucleases following endonucleolytic cleavage by Cas6[22,43,44], which supports assembly of effector complexes with varying Cas7 stoichiometries[45]. The 3′ end of the crRNA is the most exposed in these complexes due to the absence of Cas11 at this position, enabling efficient initiation of target recognition, but conformation of the crRNA in this region is identical to the other Cas7-bound crRNA 6-nt segments[14,45,46]. In the type III-Dv complex, however, the 3′ crRNA seed is held in a distinct conformation, perpendicular to the direction of the Cas7-bound crRNA segments (Fig. 1). The gating mechanism described here ensures targeting fidelity by physically preventing propagation of crRNA-target duplex hybridization in the absence of seed pairing, reminiscent of Argonaute proteins[47,48].

Type III-E single-subunit effector complexes also contain a Cas7-insertion similar to the type III-Dv complex, which has also been proposed to seed RNA target recognition[34,49]. However, since truncation of this domain does not affect in vitro or in vivo activity of type III-E effector[33], it is unlikely to play a significant role in RNA targeting, and is thus distinct from the Cas7-insertion in type III-Dv[22,47,48]. In summary, while type III effector complexes may use similar overall strategies for seeding target RNA binding (namely the 3′ location of the seed region), based on our structural and biochemical data the type III-Dv effector complex uses a distinct mechanisms whereby the 3′ end is of the crRNA is held in a conformation amenable for efficient target hybridization.

## Programmable target RNA cleavage by the type III-Dv complex

We subsequently investigated the cleavage activity of the type III-Dv complex. Incubation of the complex with a 5′-fluorescently-labeled 60-nt RNA substrate revealed cleavage of the RNA at positions 31, 37, and 43-nt from the 5′ end (Fig. 3a, Supplementary Fig. 7a). Cleavage was efficient (Fig. 3b) and metal-dependent, with optimal cleavage occurring with $Mg^{2+}$ and $Mn^{2+}$ (Supplementary Fig. 7b, c). Cleavage of the same substrate with a 3′ fluorescent label revealed only one predominant cleavage event positioned 17-nt from the label (43-nt from the 5′ end), suggesting a faster rate of cleavage at this position (Fig. 3b, Supplementary Fig. 7d). Interestingly, cleavage was observed almost immediately (Fig. 3b). The observed 6-nt spacing between cleavage products and the metal dependence has previously been observed for other type III systems[6–8]. Structural analysis of other type III systems revealed a conserved Cas7 aspartate residue in proximity of the scissile phosphate that is essential for RNA hydrolysis. However, there is currently limited detail on the mechanism of RNA cleavage, including the role of divalent cations and their placement in the active site.

To investigate the mechanism of RNA cleavage, we froze type III-Dv complexes bound to self-target RNA onto cryo-EM grids after addition of $MgCl_2$ and solved two structures in pre- and post-cleavage states at 3 Å and 3.44 Å resolution, respectively (Supplementary Fig. 6c). In our post-cleavage structure, the complex retains the 5′ end of the RNA target while releasing the 3′ end. Interestingly, Cas10 appears to be in the same conformation as the target-less complex, while the top of the complex perfectly aligns with the target bound structures (Supplementary Fig. 6d). This highlights an auto-inhibition mechanism whereby the 5′ end of a cleaved RNA target remains stably associated with the crRNA while the 3′ end dissociates, which would allow Cas10 to return to an inactive conformation that ceases cOA production and prevents initiation of host dormancy or death. Phylogenetic analyses revealed that most Cas10 subunits from type III-D systems are predicted to not have an HD domain (Makarova et al.[10]; Weigand et al.[50]). Indeed, a large truncation was observed in our type III-Dv Cas10 when compared to a type III-A Cas10 with known HD nuclease activity[13] (Supplementary Fig. 6). While an HDD sequence was located in type III-Dv Cas10, we predict the disrupted HD pocket would inactivate the nuclease activity of this subunit. We tested the ability of Cas10 within the type III-Dv effector complex to produce cOA and activate the ancillary nuclease Csx1 from *Synechocystis*[51]. As expected, target RNA, but not non-cognate RNA, activates Cas10 to produce cOA, which in turn activates the non-specific RNase activity of Csx1 to cleave a Fluorophore-Quencher reporter RNA (Supplementary Fig. 6e). Mutation of the Cas10 Palm domain catalytic residues abrogates this activity (Supplementary Fig. 6e). Therefore, we propose the differences in Cas10 structure with either the self or non-self RNA targets bound correspond to the different activation states, respectively.

Within our pre-cleavage structure, the Cas7 aspartate residues essential for catalysis are positioned between the scissile phosphate of the target RNA and the β-hairpin thumb, as observed for previous structures of type III complexes effectors[14,28,33]. These residues correspond to D26 of Cas7-Cas5-Cas11 (position 43 of the target), D33 of Cas7-2x.1 (position 37 of the target), and D246 of Cas7-2x.2 (position 31 of the target) (Fig. 3c). Within each active site, we observed unaccounted for spherical density that is positioned adjacent to both the identified aspartate residue and the scissile phosphate, which we modeled as $Mg^{2+}$ using Molecular dynamics (MD) and Quantum Mechanics/Molecular Mechanics (QM/MM) simulations (vide infra). These density features do not appear in our $Mg^{2+}$-free structures, further supporting their assignment. Individual aspartate-to-alanine mutations of these three residues were made and tested for cleavage activity (Fig. 3d, Supplementary Fig. 7e). Mutation of the predicted active site aspartate residues in the Cas7 domains successfully disrupted each cleavage event independently of the other. This feature could allow programming at these discrete and independent cleavage sites and be exploited to utilize the type III-Dv complex as a programmable sequence-specific RNase.

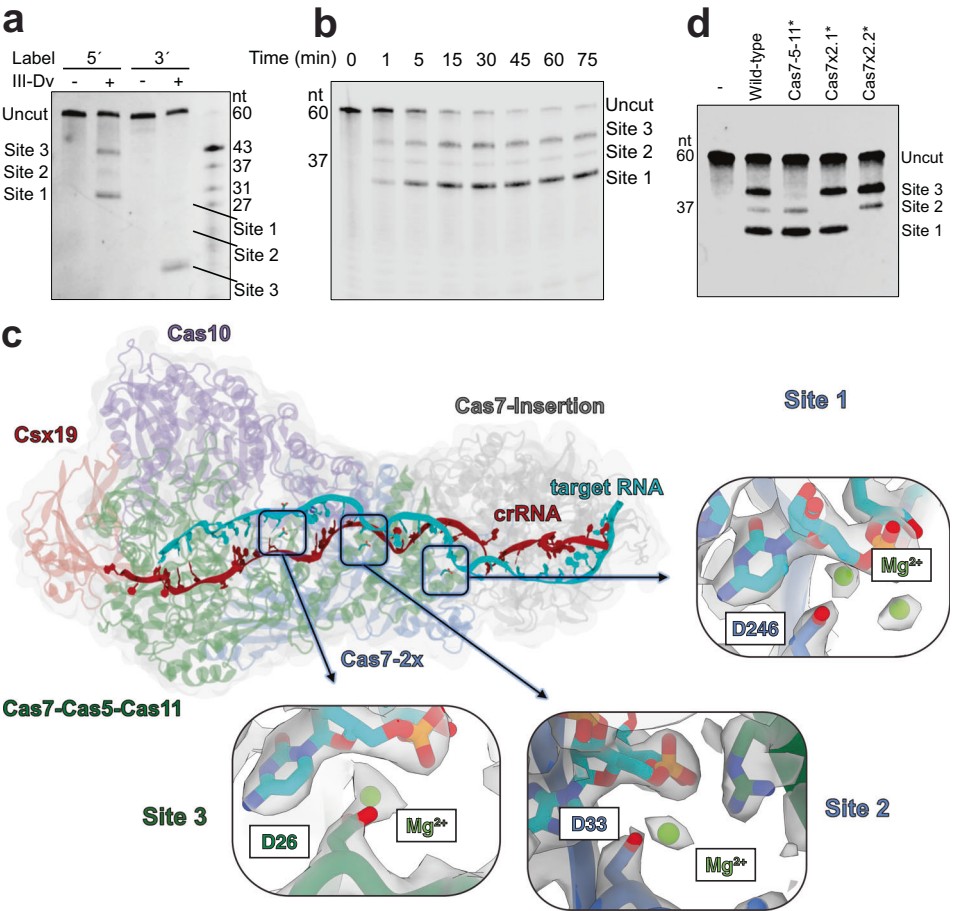

**Fig. 3 | RNA targeting by type III-Dv. a** Cleavage of the RNA target with a 5′-FAM and 3′-FAM label. Products are visualized via TBE-Urea PAGE. Representative of two independent experiments. **b** RNA cleavage time course with a 5′-IRD800-labeled RNA target across 75 min. Representative of two independent experiments. **c** 3′FAM-labeled RNA cleavage analysis after mutagenesis of the three active site aspartate residues of Cas7-Cas5-Cas11 (D26A), Cas7-2x.1 (D33A), and Cas7-2x.2 (D246A). Representative of two independent experiments. **d** Overall structure of the type III-Dv complex with potential active sites mapped. Each active site presents a spherical non-proteinaceous density, which is absent in the $Mg^{2+}$-free map and aligns well with the modeled $Mg^{2+}$ ions (green) obtained from quantum-classical simulations (details in Supplementary Fig. 8). Each $Mg^{2+}$ ion is coordinated by an aspartate residue. Source data are provided as a Source data file.

## QM/MM simulations support coordination of divalent cations by active site aspartates

We next performed classical and quantum mechanical molecular simulations to gain further insights and formulate possible hypotheses on the catalysis by each Cas7. This approach enabled us to characterize the structure and dynamics of the active site and associated metal ions, which is important for type III CRISPR-Cas complexes. Our microsecond-long classical molecular dynamics (MD) simulations allowed $Mg^{2+}$ ions to spontaneously diffuse and stably locate at the level of each active site (Supplementary Fig. 8). Specifically, two $Mg^{2+}$ ions can be accommodated within sites 1 and 3 and one $Mg^{2+}$ in site 2 (Supplementary Fig. 8a, b). QM/MM simulations were performed to characterize the metal ion coordination and revealed that, in all 3 sites, one $Mg^{2+}$ ion is coordinated by an aspartate (D246 of Cas7-2x.2 in site 1; D33 of Cas7-2x.1 in site 2; D26 of Cas7-5-11 in site 3) and the scissile phosphate from the target RNA, with water molecules saturating the metal coordination sphere (Fig. 4a, Supplementary Fig. 8c, d). We note that the placement of the diffused $Mg^{2+}$ ions is consistent with the experimental cryo-EM map (Fig. 3d). Indeed, the ions engage in coordinating the RNA backbone and the protein residues where weak density is experimentally observed. Abrogation of RNA cleavage through mutagenesis of these conserved aspartate residues indicates that this coordination is essential for RNA cleavage in all 3 sites of the type III-Dv complex, revealing a key, conserved role

for catalytic aspartates across all type III systems (Fig. 3c, Supplementary Fig. 9).

## RNA cleavage occurs through 2′-O-transphosphorylation

The general strategies for catalyzing phosphodiester bond cleavage by nucleolytic ribozymes include (1) formation of an in-line positioning of the 2′-OH nucleophile with the scissile 5′-O−P bond (α catalysis), (2) the stabilization of the negative charge on the non-bridging phosphoryl oxygens (NPOs, ß catalysis), (3) the activation of the nucleophile by a general base (γ catalysis), and (4) the facilitation of the 5′ leaving group by an acidic group (δ catalysis)[52,53]. Accordingly, the examination of each active site in the CRISPR-Cas type III-Dv complex enabled us to speculate on the role of the chemical groups in the catalysis.

The comparison of the CRISPR-Cas type III-Dv active site structures with other type III complexes reveals several similarities (Supplementary Fig. 9). All type III active sites share a catalytic aspartate crucial for RNA cleavage. Residues in active site 2 of the type III-Dv complex resemble the arrangement observed in types III-A, III-B, and III-E (site 1). Cleavage sites 3 (type III-Dv) and 2 (type III-E) both have a histidine near the RNA 2′-OH. The target RNA geometry at the catalytic sites of type III-Dv resembles that of the hammerhead, pistol, and varkud ribozymes (Fig. 4a, b, f). This geometry enables an in-line nucleophilic attack of the scissile phosphate by the 2′-OH (Supplementary Fig. 10a).

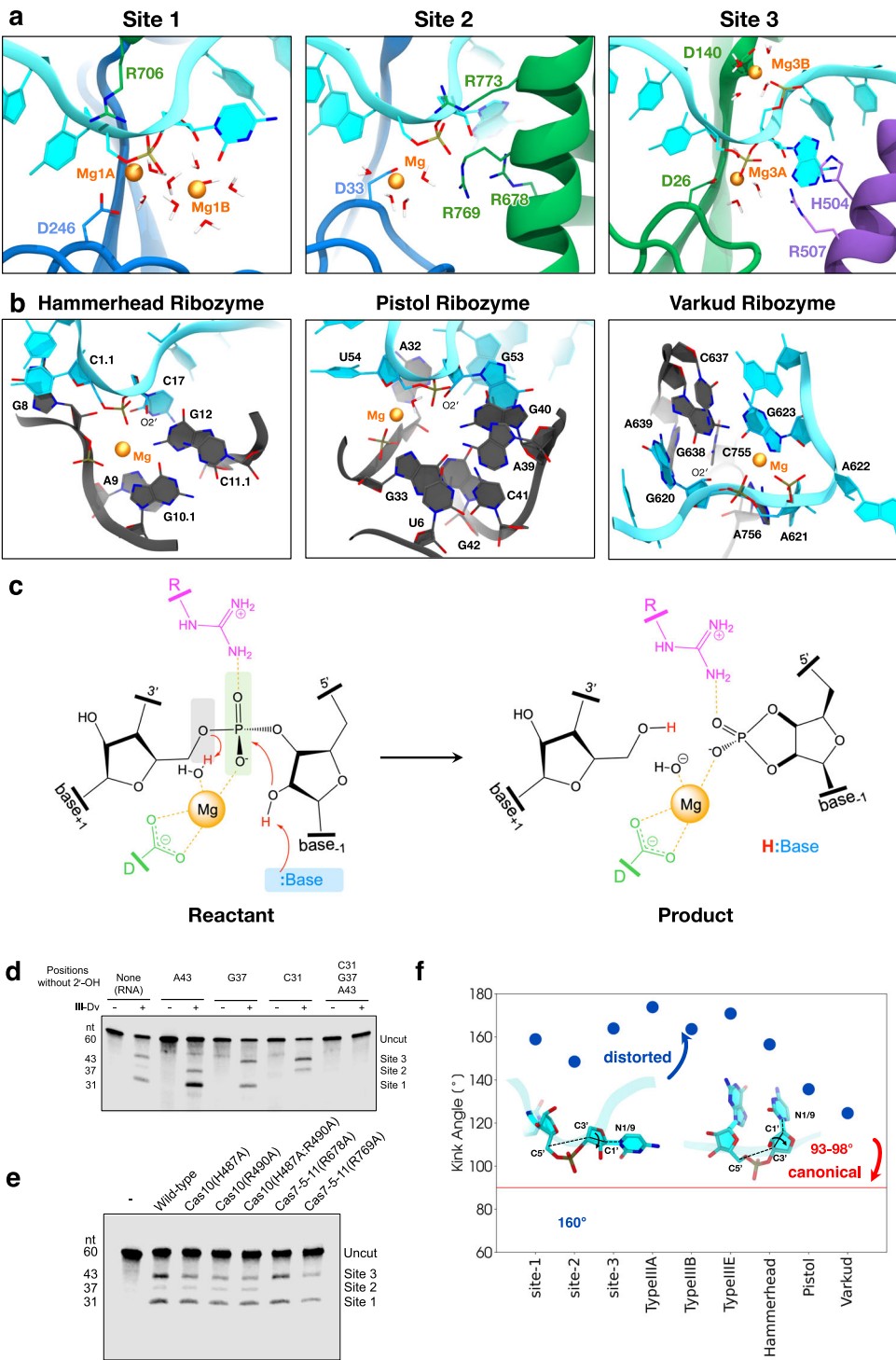

**Fig. 4 | Structural basis for target RNA cleavage. a** Active sites (sites 1, 2, and 3) of the type III-Dv CRISPR-Cas effector complex, reporting the coordination of Mg²⁺ ions and surrounding residues, from ab initio QM/MM MD simulations. **b** Structures and active sites of Hammerhead (HHr), Pistol (Psr), and Varkud (VSr) ribozymes. Shown are the representative active sites conformational state derived from MD simulations in solution. HHr crystal structure (PDB ID 2OEU[86]) and MD active state[87]; Psr crystal structure (PDB ID 5K7C[88]) and MD active state[89]; VSr dimer crystal structure (PDB ID 5V3I[90]) and MD active state[56]. **c** Schematic representation of a 2'-O-transphosphorylation via acid-base catalysis proposed for target RNA cleavage. The 2'-OH group of the adjacent base acts as a nucleophile on the scissile phosphate, aided by metal ions and nearby arginine residues. **d** 5'IRD800-labeled RNA cleavage analysis after mutagenesis of the three active sites showing deletion of nucleophilic 2'-OH at RNA positions 31, 37, and 43-nt of RNA substrate renders the substrate uncleavable at these sites. Representative of two independent experiments. **e** Effect of mutagenesis of residues in sites 2 and 3 on target RNA cleavage. Representative of two independent experiments. **f** A distortion in the target bases is reported for each site of type III-Dv complex, related type III systems (type III-A, B & E, and ribozymes. The target base flips out, positioning at a large angle from the scissile phosphate relative to A-form RNA. Pictured are bases with a canonical angle of 93–98˚ (also red line) compared to the distorted angle observed between bases in type III active sites and ribozymes. Source data are provided as a Source Data file.

This positioning of the 2′-OH nucleophile in line with the scissile 5′-O–P bond is similar to several ribozymes (Fig. 4a, b, Supplementary Fig. 10a). The variance observed in the catalysis upon replacing the metal with other similarly-sized divalent metal ions with diverse $pK_a$ values (Supplementary Fig. 10b), suggests a site-specific RNA cleavage by 2′-O-transphosphorylation[52]. In this mechanism, a general base deprotonates the 2′-OH group, activating it as a nucleophile that attacks the scissile phosphate, forming a 2′-3′-cyclic phosphate terminus and 5′ hydroxyl group as a product (Fig. 4c). Previous studies of the type III-B complex revealed the formation of 2′-3′-cyclic phosphates[4], indicating that the 2′-OH engages in RNA cleavage. To test this proposed mechanism, we performed cleavage assays on RNA substrates with the nucleophilic 2′-OH groups removed at positions 31, 37, and 43-nt of RNA substrate. Although RNA target binding was retained, replacing the 2′-OH groups with hydrogen inhibited target RNA cleavage (Fig. 4d; Supplementary Fig. 7g), which supports that the nucleophilic attack occurs by the upstream 2′-OH. This result is consistent with previous studies showing that the type III complexes can bind but do not cleave DNA substrates[4–6,24].

## Structural basis of acid-base catalysis and resemblance to ribozymes

Based on structural data, Suslov et al. identified a common active site motif—viz., the "L-platform"—in the Varkud Satellite (VSr), Hairpin (HPr), and Hammerhead (HHr) ribozymes[54,55]. In this platform, an L-anchor nucleobase positions a conserved G, serving as the general base, to activate the 2′-OH nucleophile. The L-pocket creates a divalent metal ion binding site, distinguishing two catalytic paradigms: (i) the "G + A (guanine + alanine)" paradigm [e.g., Twister ribozymes (Twr), HPr, VSr] that exclusively employs nucleobase functional groups, and (ii) the "G + M⁺ (guanine + metal)" paradigm [e.g., Pistol ribozyme (Psr), 8–17 DNAzyme, HHr), which structures the L-pocket to recruit a divalent metal ion that actively participates in the chemical reaction[55]. In each active site of the type III-Dv complex, the target RNA arrangement is similar to the above-mentioned ribozymes, and the presence of divalent metal ions is crucial for catalysis. This suggests that, in principle, the type III-Dv active sites align with the "G + M⁺" paradigm. Each active site includes metal ions without causing significant structural changes (RMSD < 1 Å between structures with and without $Mg^{2+}$), and these metal ions actively contribute to the chemical reaction. This is different from the VSr ("G + A" paradigm), where $Mg^{2+}$ plays an organizational role rather than directly engaging in catalysis[56].

Divalent ions are reported to aid the catalysis in RNA and DNA in various ways[57–59], such as by stabilizing the NPOs or the O5′ leaving group through a direct electrostatic effect, by acting as a general acid, or by tuning the $pK_a$ of functional groups like 2′−OH[60]. Considering this knowledge, we examined the active sites of our type III-Dv complex to hypothesize the possible catalytic scenarios.

In Site 1, two $Mg^{2+}$ ions (Mg1A, Mg1B) coordinate the scissile phosphate, and no protein residues or nucleobases can be identified for acid-base catalysis (Supplementary Fig. 10c, d). Hence, the first hydrated $Mg^{2+}$ ion (Mg1A) may act as an acid, supported by a reduced catalytic ability detected in the presence of higher $pK_a$ $Ca^{2+}$ ions, and an enhancement of the activity with lower $pK_a$ $Ni^{2+}$ ions (Supplementary Figs. 7b, 10b). Mg1A coordinates the O5′, suggesting a δ catalysis, assisted by the R706(Cas7-Cas5-Cas11)−NPO coordination, which is also an indication of ß/γ″ catalysis. In this configuration, the first shell water molecule, bound to Mg2B, donates a hydrogen bond to the 2′-OH, indicating γ′ catalysis, while Mg2B−NPO coordination implies γ″ catalysis, collectively shifting the $pK_a$ of the 2′-OH (Supplementary Fig. 10c−e).

Site 2 shows three arginine residues (R678, R769, R773 of Cas7-Cas5-Cas11) and one metal ion. The hydrated $Mg^{2+}$ may act as an acid, as we noticed reduced catalytic ability with higher $pK_a$ $Ca^{2+}$ ions, while an increase in the catalysis is observed with lower $pK_a$ $Ni^{2+}$ ions

(Supplementary Figs. 7b, 10b). The interaction of the NPOs with R769 and R773 suggests their role as ß or γ″ catalysts, stabilizing negative charge at the transition state or preventing hydrogen bond formation between the 2′-OH and the NPOs. In addition, R678 closely interacts with the nucleophilic 2′-OH, donating two hydrogen bonds via its guanidinium group, suggesting γ′ catalysis (Supplementary Fig. 10c−e). These hypotheses find support in the loss of catalytic activity at Site 2 upon R678A and R769A mutations (Fig. 4e).

Site 3 consists of two $Mg^{2+}$ ions and two protein residues (H487 and R490 of Cas10). The partial reduction in cleavage at site 3 in the H487A and R490A mutants of the type III-Dv complex (Fig. 4d), both individually and collectively, suggests that these residues may not act as a base but could engage in a 2°/3° γ catalysis. The direct interaction between Mg3A and the NPO suggests its role as ß or γ″ catalysts. However, the hydrogen bond donation by the first shell water molecule to the 2′-OH indicates γ′ catalysis. Interestingly, the D140 of Cas7-Cas5-Cas11 residue coordinates a second $Mg^{2+}$ ion (Mg3B) on top of the catalytic center. This Mg3B could be critical for a proton transfer to the O5′, through its first shell water molecule, hydrogen bonding the $C44_{O2'}$. In summary, our data provide insights into the target RNA cleavage via a 2′-O-transphosphorylation mechanism, which is likely conserved across all type III complexes.

## Discussion

Recent metagenomic and biochemical analysis of the type III-E single protein effector has led to hypotheses about how these systems evolved from the multi-subunit type III effectors[8–10,25]. Our structures reveal how the type III-Dv system utilizes subunit fusions while maintaining the domain organization from type III-D1[10,39]. Many of the subunits individually align well to the type III-A and type III-B subunits, but the type III-Dv complex does not vary in crRNA length or complex size. The Cas7-insertion subunit pulls the 3′ end of the crRNA away from the complex, exposing a 6-nt seed which initiates crRNA-target hybridization. While types III-Dv and III-E appear to use different mechanisms for seed base-pairing, both complexes contain similar Cas7-insertions, which supports the evolutionary relationship between the two complexes[9,10]. We also reveal the presence of the unique and uncharacterized subunit Csx19, which nestles between Cas10 and Cas5 at the base of the complex and is required for complex assembly or stability. In addition, the presence of subunit fusions and the Cas7-insertion subunit sandwiches the type III-Dv complex between multi- and single-subunit type III complexes in the evolutionary progression of type III CRISPR systems[9,10].

In our structures with different self and non-self RNA targets bound, we observe conformational changes to Cas10 that are consistent with activation of the palm domain[13] (Supplementary Fig. 6). Unlike type III-E complexes, type III-Dv retains a GGDD motif and an active site that is involved in cyclic oligoadenylate (cOA) signaling (Supplementary Fig. 6). Future studies are necessary to fully understand the mechanism of Cas10 activation and cOA production in this system.

Type III CRISPR systems target single-stranded RNA, which makes them powerful post-transcriptional silencers of phage RNA[4,61]. Previous studies illustrated a 6-nt ssRNA cleavage periodicity by the III-A, III-B, and III-E effectors[6,9,20,25]. Here, we show the in vitro activity of the type III-Dv effector complex on a ssRNA target and reveal three active sites in the Cas7-Cas5-Cas11 and Cas7-2x fusion subunits. The type III-Dv complex cleaves on the scale of minutes and each site can be independently controlled for cleavage, emphasizing the potential use of the type III-Dv complex as a programmable RNA endonuclease.

Our work builds upon over a decade of previous studies which have proposed similar catalytic mechanisms. It has been previously shown that type III-B RNA cleavage products had 2′-3′ cyclic phosphate ends, suggesting that the cleavage event was divalent metal ion-dependent and RNA-catalyzed[4,8]. Various studies have also shown that

removal of 2′OH on the RNA target at specific positions abrogates cleavage[26,27]. Our work expands on the existing literature by providing a structure of an active type III effector complex where the locations of $Mg^{2+}$ ions essential for catalysis are resolved, which subsequently enabled comprehensive molecular dynamics simulations and a more comprehensive model of catalysis.

Our consideration of the acid-base chemistry of RNA cleavage by the type III-Dv CRISPR-Cas complex revealed that the positioning of the 2′OH juxtaposed next to the scissile phosphate in all three active sites of this complex mirrors active sites of other type III complexes and shares similarities to certain ribozymes. The type III-Dv pre-cleavage complex structure also illustrates how the notable catalytic aspartate residues in type III complexes coordinate metal ions essential for catalysis either through an active role as an acid or by stabilizing non-bridging phosphoryl groups. We identified the role of R678 in Cas7-5-11 as a catalytic base in site 2, a mechanism likely conserved across type III-A and III-B systems. The flipped-out target base is indeed a consistent feature of the type III CRISPR-Cas complexes, positioned at a ~160° angle from the scissile phosphate, creating a geometry favorable for target RNA cleavage. While every class 1 CRISPR effector visualized to date features the flipping of a target base at every 6th position[26,28,33,35,37–42], only type III effector complexes perform RNA target cleavage[62]. Since a type III-like ancestor was the likely progenitor of other class 1 CRISPR effectors[63], the flipped target base may be a vestigial evolutionary remnant that has persisted even when no longer required for nucleophilic attack.

## Methods

### Plasmids and oligonucleotides
Refer to Supplementary Tables 3 and 4 for lists of all plasmids and oligonucleotides used in this study.

### Culture conditions
Refer to Supplementary Table 5 for a list of all strains used in this study.

Unless otherwise noted, *Escherichia coli* strains were grown at 37 °C in Lysogeny Broth (LB), or on LB-agar (LBA) plates with 1.5% (w/v) agar. Media were supplemented with antibiotics when required as follows: chloramphenicol (Cm; 25 μg/mL), and kanamycin (Km; 50 μg/mL).

### Construction of plasmids
A plasmid (pPF2434) for expression of Cas10, Cas7-5-11, Cas7-2x, Csx19, and Cas7-insert was constructed by PCR-amplifying their genes (primers PF4851+ PF4852) using *Synechocystis* genomic DNA as template and cloning the product into pRSF-1b via KpnI and PstI restriction sites. The *cas10* gene was cloned to incorporate an N-terminal His6 tag followed by TEV protease recognition sequence.

A plasmid (pPF2441) for expression of the first spacer (5'-TGTAG-TAGAACCAATCGGGGTCGTCAATAACTCCCG-3') and flanking repeat sequences (5'-GTTCAACACCCTCTTTTCCCCGTCAGGGGACTGAAAC-3') from the type III-Dv associated CRISPR array was constructed by PCR-amplifying this region from *Synechocystis* genomic DNA (primers PF4847+ PF4848) and cloning the product into pACYCDuet-1 via NdeI and KpnI restriction sites. A plasmid (pPF2442) was constructed for expression of Cas6-2a with the first spacer and flanking repeat sequences by PCR-amplifying *cas6-2a* (primers PF4849+ PF4850) using *Synechocystis* genomic DNA as template and cloning the product into pPF2441 via NcoI and BamHI restriction sites.

Plasmids pPF3085, pPF3086, pPF3089, pPF3205, pPF3336, PF3436, pPF3519, pPF3521, pPF3522 and pPF3636 are for expression of mutants Cas7-2x(D29A, D31A, D33A), Cas7-2x(D241A, D246A), ΔCsx19 (nonsense mutation), Cas7-5-11(D26A), Cas10(D308A; D309A), Cas10(D487A), Cas10(R490A), Cas10(D487A; R490A), Cas7-5-11(D678A) and Cas7-5-11(D769A), respectively. Plasmids pPF3085, pPF3086, pPF3089, pPF3205, pPF3336, pPF3436, pPF3519, pPF3521, and pPF3522

were constructed by site-directed mutagenesis through amplifying plasmid pPF2434 with primers PF5991+PF5992, PF5993+PF5994, PF6281+PF6282, PF6423+PF6424, PF6780+PF6781, PF6983+PF6984, PF6994+PF6995, PF6998+PF6999 and PF7000+PF7001, respectively. Plasmid pPF3636 was constructed by site-directed mutagenesis through amplifying plasmid pPF3436 with primers PF7305+PF7306. Each reaction was treated with DpnI to remove PCR template, and Gibson assembly was used to ligate the PCR product into the mutated plasmid. A plasmid (pPF2932) for expression of Csx1 was constructed by PCR-amplifying *csx1* (primers PF5629+PF5630) using *Synechocystis* genomic DNA as template and PCR-amplifying plasmid pPF2500 (PF5629+5630). Each reaction was treated with DpnI, and Gibson assembly was used to ligate the fragments. The *csx1* gene was cloned to incorporate N-terminal His$_8$ and SUMO tags.

### Expression and purification of Csx1
Csx1 with N-terminal SUMO and His$_8$ was expressed in LOBSTR cells containing plasmid pPF2932. Five hundred mL cultures were induced with 0.5 mM IPTG at $OD_{600} = 0.6$ and grown overnight at 18 °C. Cells were harvested at $10,000 \times g$ for 10 min. The cell pellet was resuspended in 20 mL of lysis buffer supplemented with 0.02 mg/mL DNaseI and cOmplete EDTA free protease inhibitor (Roche). Cells were lysed by a French pressure cell press (American Industry Company) at 10,000 psi, and the lysate was clarified by centrifugation at $15,000 \times g$ for 15 min. The lysate was applied to a HisTrap affinity column (GE Healthcare) equilibrated in lysate buffer and eluted using a gradient against lysate buffer containing 500 mM imidazole. The fractions containing the Csx1 were pooled, treated with Ulp1_R3 protease and incubated at 4 °C during overnight dialysis in SEC buffer (10 mM HEPES-NaOH, pH 7.5, 100 mM KCl, 5% Glycerol, 1 mM DTT). The liberated His$_6$-SUMO tag and non-specific *E. coli* proteins were removed using a second HisTrap affinity column and the flow through was collected. The sample was concentrated with a centrifugal concentrator (Amicon; 100 kDa molecular weight cut off (MWCO)) and further purified by SEC on a HiLoad 16/600 Superdex 200 (GE Healthcare) column equilibrated in SEC Buffer. Purified Csx1 was concentrated to 1.0 mg/mL using a centrifugal concentrator (Amicon; 100 kDa MWCO), aliquoted, and stored at −80 °C.

### Expression and Purification of type III-Dv effector complex
Type III-Dv complex with N-terminal His$_6$-TEV-Cas10 was expressed in LOBSTR cells containing plasmids pPF2434 and pPF2442. Five hundred mL cultures were induced with 0.5 mM IPTG at $OD_{600} = 0.6$ and grown overnight at 18 °C. Cells were harvested at $10,000 \times g$ for 10 min. The cell pellet was resuspended in 20 mL of lysis buffer (50 mM HEPES-NaOH, pH 7.5, 300 mM KCl, 5% Glycerol, 1 mM DTT, 10 mM imidazole) supplemented with 0.02 mg/mL DNaseI and cOmplete EDTA free protease inhibitor (Roche). Cells were lysed by a French pressure cell press (American Industry Company) at 10,000 psi, and the lysate was clarified by centrifugation at $15,000 \times g$ for 15 min. The lysate was applied to a HisTrap affinity column (GE Healthcare) equilibrated in lysate buffer and eluted using a gradient against lysate buffer containing 500 mM imidazole. The fractions containing the type III-Dv complex were pooled, treated with TEV protease and incubated at 4 °C during overnight dialysis in SEC buffer (10 mM HEPES-NaOH, pH 7.5, 100 mM KCl, 5% Glycerol, 1 mM DTT). The sample was applied to a second HisTrap column; however, due to inefficient TEV cleavage, the complex unexpectedly bound the column and eluted with high imidazole. The complex was further purified by size exclusion chromatography (SEC) on a HiLoad 16/600 Superdex 200 column (GE Healthcare) equilibrated in SEC Buffer. Mutant type III-Dv complexes were similarly expressed and purified, except TEV protease was omitted. Purified complexes were typically concentrated to 1.5 mg/mL using a centrifugal concentrator (Amicon; 100 kDa MWCO), aliquoted, and stored at −80 °C (Supplementary Fig. 2).

## Type III-Dv activation of Csx1 assay

Type III-Dv activation assays were conducted in 60 μL reaction volumes containing 100 nM purified type III-Dv effector complex (wild-type or Cas10-Palm mutant), 100 nM purified Csx1, 333 nM RNA substrate (target RNA, PF5855; non-cognate RNA, PF4100), and 150 nM fluorescent reporter RNA (PF7782) in final buffer conditions of 12.5 mM Tris-HCl (pH 8.0), 2 mM HEPES-NaOH (pH 7.5), 17 mM KCl, 10% glycerol, 1 mM DTT, 10 mM $MgCl_2$, 0.25 mM ATP, 20 mM NaCl. Reactions were incubated at 25 °C for 5 min. Fluorescence signal from cleavage of the fluorescent reporter RNA (excitation at 480 nm, emission at 530 nm) was measured in a Victor Nivo alpha F (Revvity) multimodal plate reader. The fluorescent reporter RNA substrate for Csx1 was ordered as ssRNA (IDT) labeled with 5' 6-FAM and 3' IOWA Black® dark quencher. All targets used for cleavage are listed in Supplementary Table 4.

## Native mass spectrometry

5 μL aliquots of the CRISPR complex solution were buffer exchanged into 100 mM ammonium acetate using Biospin P-6 gel columns (Bio-Rad Laboratories Inc., Hercules, CA) prior to native mass spectrometry. Samples were loaded into gold/palladium-coated borosilicate static emitters and subjected to electrospray ionization using a source voltage of 1.0–1.3 kV and analyzed in the positive ion mode on a Thermo Scientific Q Exactive Plus UHMR Orbitrap mass spectrometer (Bremen, Germany). Subcomplexes and ejected subunits were produced and measured via quadrupole isolation of the intact complex charge envelope, followed by higher-energy collisional dissociation (HCD) using 290 eV normalized collision energy (NCE). Ion optics and trapping gas pressure were tuned for the transmission and detection of each set of analytes, including the intact complex, subcomplexes, and ejected subunit ions. Native mass spectra were collected by averaging 500 microscans at a resolution of 1625 at $m/z$ 200. Spectra were deconvoluted using UniDec.

To confirm the presence of individual subunits, the CRISPR complex was subjected to denaturing LC-MS analysis. Denaturing liquid chromatography mass spectrometry (LC-MS) was performed on a Dionex UltiMate 3000 nanoLC system coupled to a Thermo Orbitrap Fusion Lumos Tribrid mass spectrometer (San Jose, CA). The trap column (3 cm) and analytical column (30 cm) were packed in-house with polymeric reverse-phase (PLRP) packing material. Approximately 80 ng of the CRISPR complex were injected and subjected to reverse-phase chromatography, utilizing water with 0.1% formic acid as mobile phase A (MPA), and acetonitrile with 0.1% formic acid as mobile phase B (MPB). Forward trapping occurred for 5 min at 2% MPB at a flow rate of 5 μL/min at the trap column. Elution onto the analytical column (at 0.3 μL/min) occurred by increasing MPB to 10% over a 3-min gradient followed by an increase to 35% MPB over 32 min. Mass spectra were collected at a resolution of 15,000 at $m/z$ 200, using 5 microscans and an AGC target of 1E6. Spectra were manually averaged over each subunit elution period and deconvoluted with UniDec.

## RNA cleavage by the type III-Dv effector complex

RNA targets for testing cleavage were either ordered as fluorescently labeled ssRNA (IDT) with either 5' IRD800, 5' 6-FAM, or 3' 6-FAM label label. All targets used for cleavage are listed in Supplementary Table 2. RNA cleavage assays were conducted in 5 μL of reaction typically containing 200 nM purified type III-Dv effector complex, 100 nM RNA substrate in final buffer conditions of 6 mM HEPES-NaOH, pH 7.5, 60 mM KCl, 5 mM $MgCl_2$, 3% glycerol, 1 mM DTT. Reactions were incubated at 37 °C for 30 min or as indicated. Reactions were stopped by adding 1 μL 6 M guanidinium thiocyanate and 6 μL 2x RNA loading dye. Samples were heated for 5 min at 95 °C and immediately placed on ice for 3 min. Samples were analyzed by 1× TBE, 15% acrylamide, 8M urea denaturing PAGE (Thermo Fisher). Fluorescent probes were imaged using the Odyssey Fc imaging system (LICOR).

## Electrophoretic mobility shift assays

RNA targets for EMSAs were ordered as fluorescently labeled ssRNA (IDT) with either 5' IRD800 or 5' 6-FAM label. All targets used are listed in Supplementary Table 2. EMSAs were conducted in 5 μL of reaction containing 100 nM purified type III-Dv effector complex, 100 nM RNA substrate in final buffer conditions of 6 mM HEPES-NaOH, pH 7.5, 60 mM KCl, 5 mM EDTA, 3% glycerol, 1 mM DTT. Reactions were incubated at 37 °C for 30 min. Samples were separated on 4% poly-acrylamide (19:1 acrylamide:bisacrylamide) native gel containing 0.5× TBE at 4 °C. The fluorescent probe was imaged using the Odyssey Fc imaging system.

## Cryo-EM grid preparation and data collection

Fully assembled type III-Dv binary complex was diluted to a concentration of 0.3 mg/mL in SEC buffer before 2.5 μL of sample was added to a quantifoil 1.2/1.3 grid that was glow discharged for 1 min. Sample was applied to the grid in an FEI Vitrobot MarkIV kept at 100% humidity and 4 °C before blotting for 5.5 s with a force of 0. For the RNA target-bound complex, non-self RNA target (PF5855) was mixed with the binary complex with a 2:1 molar ratio of RNA:binary complex for 30 min at 30 °C in SEC buffer to a final non-self-target-bound complex concentration of 0.3 mg/mL. Pre- and post-cleavage complexes were captured by mixing self-target RNA and binary complex in a 2:1 molar ratio, followed by addition of $MgCl_2$ in SEC buffer to a final concentration of 10 mM of $Mg^{2+}$ and 0.3 mg/mL of self-target-bound complex. Grids of the non-self- and self-target-bound complexes were frozen with identical conditions to the binary complex. Grids were loaded to an FEI Titan Krios (Sauer Structural Biology Lab, University of Texas at Austin) operating at 300kV. Images were taken at a pixel size of 0.81 Å/pixel with a dose rate of 16.2 $e^-/Å^2/s$ for 5 s for the binary and non-self-target-bound complexes and at a pixel size of 0.8332 Å/pixel with a dose rate of 20.2 $e^-/Å^2/s$ for 4 s using a Gatan K3 direct electron detector, giving a final dosage of ~80.5 $e^-/Å^2$ for all datasets. A Gatan Bio-continuum operating at a 20 eV slit was also used for the self-target-bound dataset. Data collection was automated using SerialEM v3.8 using a defocus range of −1.2 to −2.2 μm.

## Cryo-EM data processing

Movies from the Gatan K3 were motion corrected using motioncor2, and corrected micrographs were uploaded to cryoSPARC v2[64]. After CTF correction, initial templates for template-based picking were generated using a blob picker and 2D classification. Template-based particle picking resulted in ~1.85 million particles (binary complex) and ~1.92 million particles (target-bound complex) being picked.

Processing the dataset for the binary complex was started with one round of 2D classification, sorting out particles to a new subset of ~926k particles. Ab initio reconstruction and subsequent hetero-geneous refinement with four classes was utilized and ~649k particles were selected from one of the classes. Particles were then split by exposure groups before performing a final non-uniform (NU) refinement[65], yielding a final map at 2.5 Å resolution. Using a mask generated in ChimeraX[66,67] around the Cas7-insertion portion of the map, we reconstructed a map of this region at 2.5 Å resolution using local refinement in cryoSPARC. The two maps were stitched into a composite map using the vop maximum command in ChimeraX v1.4.

For the target-bound complex, ~1.92 million particles were input into 2D classification and filtering, sorting out particles to a new subset of ~1.07 million particles. This new subset was then input into ab initio reconstruction and heterogeneous refinement on cryoSPARC v2 with four classes and filtered out ~453k particles to a new subset of ~614k particles[64]. These particles were split by exposure groups before performing NU refinement with identical settings to the final NU refinement in the binary complex dataset[65]. The full complex model was refined identical to that of the binary complex. This refinement yielded a 2.8 Å resolution structure from ~610k particles. To refine the density

for the Cas7-insertion subunit, we generated a mask in ChimeraX and performed local refinement on the ~610k particle set and reconstructed a 2.7 Å resolution map. A composite map was generated in ChimeraX using the vop maximum command.

The self-target-bound complex was processed similarly to the other two datasets. After pre-processing, ~1.28 million particle picks were filtered with 2D classification to a subset of ~690k particles. We then generated 3 models using ab initio reconstruction and heterogeneous refinement using cryoSPARC v3 and filtered the subset to ~486k particles. We input this particle set into 3D classification with 10 classes to generate a final particle set of ~182k particles and reconstructed a 3.01 Å map using non-uniform refinement of the self-target- and $Mg^{2+}$-bound complex in a pre-cleavage state. To generate the post-cleavage structure, we utilized 3D variability analysis with 10 clusters on the ~486k particle subset. We reconstructed a 3.44 Å map using non-uniform refinement with a subset of ~41k particles from one of the 3D variability clusters.

### In silico subunit modeling and refinement
The subunit models were generated using Alphafold 2 using the monomer model preset and fit into the map using Namdinator and ISOLDE[68–70]. The reduced database precision was used for the multiple sequence alignment. The AF2 job run included a relaxation step, resulting in both relaxed and unrelaxed models. The model of the full complex was refined using Phenix real-space-refinement[71] using the model from ISOLDE as a reference turning off secondary structure restraints, NCS restraints, and local grid search. Rotamer outliers were adjusted using Coot[72].

### All-atom MD simulations
MD simulations were performed by using the Amber ff19SB force field for the protein, including the OL3 corrections for RNA molecules[73,74]. The ternary complexes were subjected to energy minimization in vacuum for 1000 steps, using a steepest descent and a conjugate gradient algorithm. The minimized structures were solvated in explicit water, obtaining a cubic periodic box extending 12 Å from the surface of the protein. The water molecules were described using the water model-TIP3P. The solvated systems were neutralized by adding 83 $Mg^{2+}$ and 91 Cl- ions to neutralize and mimic the experimental concentration of 10 mM[75,76]. While the 12-6-4 potential for divalent metals reported a good agreement in infinitely diluted systems[75,76], the state-of-the-art by Panteva et al. provide a better description of the metal ions interacting with RNA[77]. A particle mesh Ewald method with 12 Å cutoff was used to handle the electrostatics of the systems[78]. All the systems were then equilibrated in NVT ensemble for 500ps and in NPT ensemble for 1 ns at 1 atm pressure and 300K, before the production run. A time step of 2fs was used and the coordinates were saved at every 2ps interval. Subsequently three independent runs of ~1 microsecond MD simulations were performed.

### Quantum mechanics/molecular mechanics (QM/MM) simulations
QM/MM simulations were carried out on three catalytic sites of type IIID CRISPR-Cas effector complex. In the site-1, the QM part included two $Mg^{2+}$ ions and its coordination residue (D246 (Cas7-2x.2), C31, C32 (scissile phosphate) and eight water molecules), neighboring residues R706 (Cas7-Cas5-Cas11). In the site-2, QM part included $Mg^{2+}$ ions and its coordination residue (D33 (Cas7-2x.1), G387, G38 (scissile phosphate) and four water molecules), neighboring residues R679, R779, and R779 (Cas7-Cas5-Cas11). In the site-3, QM part included two $Mg^{2+}$ ions and its coordination residue (D26, D140 (Cas7-Cas5-Cas11), C44, A43, and seven water molecules) and neighboring residues H504 & R507 (Cas10). For all the systems, capping hydrogen atoms were used to saturate the valence of the terminal QM part, resulting in a total of 106, 128, and 135 QM atoms for the site-1, site-2, and site-3 catalytic

pockets, respectively. The QM atoms were described at the QM DFT/BLYP level[79,80], while the remaining MM part was treated using the classical force field reported above. QM/MM simulations have been performed using the CPMD code. The wave functions were expanded in a plane wave basis set up to a cutoff of 75 Ry in a QM cell of dimensions ~22*26*22 $Å^3$, ~23*26*25 $Å^3$ & ~25*27*26 $Å^3$, respectively, for site-1, site-2 & site-3. The interactions between the valence electrons and ionic cores were described with norm-conserving Martins-Troullier pseudopotentials[81]. The QM part was treated as an isolated system, and electrostatic interactions between periodic images were decoupled by the scheme of Tuckerman[82]. Notably, a rigorous Hamiltonian treatment of the electrostatic interaction between the QM and MM regions was used[83]. The QM/MM protocol consisted of an initial optimization of the wavefunction, followed by ~6 ps of careful equilibration carried out with Born–Oppenheimer MD in the canonical (NVT) ensemble using an integration time step of 20 au (~0.48 fs). The temperatures of the QM and MM subsystems were kept constant at 300 K using a Nosé-Hoover thermostat[84,85]. After this initial phase, ~20 ps of Car-Parrinello QM/MM simulations were carried out with a time step of 5 au (~0.12 fs) and a fictitious electron mass of 600 au[85].

### Reporting summary
Further information on research design is available in the Nature Portfolio Reporting Summary linked to this article.

### Data availability
The cryo-EM structures and associated atomic coordinates for apo-type III-D effector (crRNA-bound), type III-D effector bound to non-self target RNA, type III-D effector with self-target RNA in a pre-cleavage state, and type III-D effector with self-target RNA in a post-cleavage state have been deposited into the Electron Microscopy Data Bank and Protein Data Bank with accession codes EMD-40248, EMD-40249, EMD-40250, EMD-40251, and PDB 8S9T, PDB 8S9U, PDB 8S9V, and PDB 8S9X, respectively. The apo-type III-D effector consensus map EMD-40276 and local refinement map EMD-40296; and the type III-D effector bound to non-self target RNA consensus map EMD-40298 and local refinement map EMD-40297 have also been deposited into the Electron Microscopy Data Bank. All materials and data are available upon request from the corresponding authors Giulia Palermo (gpalermo@engr.ucr.edu), Robert D. Fagerlund (robert.fagerlund@otago.ac.nz), and David W. Taylor (dtaylor@utexas.edu). Source data are provided with this paper.

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

## Acknowledgements

We thank A. Brilot for expert cryo-EM assistance; members of the Taylor, Brodbelt, Palermo, and Fineran labs for helpful discussions. Data were collected at the Sauer Structural Biology Laboratory at the University of Texas at Austin. This work was supported in part by National Institute of General Medical Sciences (NIGMS) of the National Institutes of Health (NIH) R35GM138348 (to D.W.T.) and R01GM141329 (to G.P.), National Cancer Institute of the NIH F31CA257404 (to L.A.M.) and R35GM139658 (to J.S.B.), Welch Foundation Research Grant F-1938 (to D.W.T.) and F-1155 (to J.S.B.), National Science Foundation (NSF) CHE-2144823 (to G.P.), a Marsden Fund Fast-Start Grant (to R.D.F.) from the Royal Society of New Zealand (RSNZ), the Marsden Fund (to P.C.F.), Bioprotection Aotearoa (Tertiary Education Commission, NZ) (to R.D.F. and P.C.F.). The content is solely the responsibility of the authors and does not necessarily represent the official views of the National Institutes of Health.

## Author contributions

R.D.F., P.C.F., and D.W.T. conceived the study. R.D.F., P.C.F., D.W.T., E.A.S., and J.P.K.B. designed the experiments. E.A.S. performed cryo-EM, structure determination, and modeling with assistance from J.P.K.B. M.A. performed computational simulations with supervision from G.P. L.A.M. and J.N.W. performed mass spectrometry supervised by J.S.B. C.L.M. performed in silico structure prediction. T.L.D. performed initial cleavage experiments. R.D.F. performed cloning, expression, and purification of the complexes, EMSAs and final cleavage experiments. All

authors analyzed and interpreted the data. E.A.S., M.A., G.P., P.C.F., R.D.F., and D.W.T. wrote the manuscript with input from all authors. J.S.B., G.P., P.C.F., R.D.F., and D.W.T. supervised and obtained funding for the work.

## Competing interests

The University of Otago and the Board of Regents for the University at Texas have filed a patent application related to this work titled "Type III-D CRISPR-Cas System and Uses Thereof" (PCT/NZ2023/050059, filed 13 June 2023), where R.D.F., P.C.F., E.A.S., J.P.K.B., and D.W.T. are inventors. All other authors declare that they have no competing interests.
