## [Peer Review File · Nature Communications]

RNA targeting and cleavage by the type III-Dv CRISPR effector complexEditorial Note: This manuscript has been previously reviewed at another journal that is not operating a transparent peer review scheme. This document only contains reviewer comments and rebuttal letters for versions considered at *Nature Communications*.

REVIEWER COMMENTS

Reviewer #1 (Remarks to the Author):

The manuscript by Schwartz et al. determined a series of CryoEM structures (+/- bound target RNA and pre- and post- target RNA cleavage) of a newly identified and biologically rare, type III-Dv complex from *Synechocystis*. Previously, the Koonin lab had identified this III-D complex and proposed to be a possible evolutionary intermediate between multi-subunit type III crRNP complexes and the recently characterized single-subunit III-E complex (but it may simply be a highly divergent and recently evolved type III system). The complex has multiple domain fusions like the type III-E complex but unlike type III-E, has Cas10, Cas5 and Csx19 subunits and lacks TPR-CHAT protease and other III-E-specific subunits. As a major focus of the study, the authors focused on determining the mechanism of type III target RNA cleavage using a combination of cryo-EM, molecular dynamics simulations, and biochemical assays. More specifically, they sought to understand how each of the three catalytic sites responsible for target RNA cleavage carry phosphodiester bond cleavage reactions and modeled predicted catalytic Mg ions examined turnover of the cleaved target RNA.

Comments:

1) No information is presented on the *in vivo* function of the complex (even in a heterologous *E. coli* host if the native organism is not genetically amenable) to excite us about the observed structural features presented. Importantly, this complex has a Cas10 subunit which in other Type III systems is capable of ssDNA cleavage (via HD domain) and/or generating cyclic oligoadenylate signaling molecules using ATP (via Palm domain). The authors now mention that a catalytically active PALM active site (GGDD) is predicted in the Cas10 but there is still no description on if the Cas10 is predicted to be catalytically active or inactive for ssDNase activity based on amino acid sequence features (e.g. hallmark HD motif, etc.; this should be clearly noted). There is no description or experimental investigation on Cas10 from this complex.

While we agree that a detailed structural and mechanistic understanding of cOA signalling by this complex is outside of the scope of this manuscript, the manuscript would be significantly strengthened if the authors experimentally demonstrated/addressed that this complex is capable of cOA signalling and/or ssDNase activity. Experimentally evaluating/demonstrating these key predicted Cas10 activities is key to our fuller understanding of the proposed evolutionary link between this system and other type III systems. Moreover, this would provide foundational knowledge toward future work aimed ultimately at gaining detailed insight into these important activities that likely provide immunity against the viruses

(The Cas7-directed RNase activity alone is likely insufficient for immunity and clearance of a DNA virus or plasmid).

2) It seems illogical to be describing conformational changes in Cas10 required for “activation” (extended figure 6) if there are no activities for this protein being examined (or is the conformational change of Cas10 in this system essential for target RNA cleavage? If so, the evidence should be explicitly described).

3) Line 205: reference #8 should be included in addition to refs 6 and 7. Line 343: Likewise, ref 8 should be included.

Reviewer #2 (Remarks to the Author):

The revised manuscript presents an improved version of the initial submission. A few points listed below still need clarification or improvement before publishing.

1. The paragraph describing the evolutionary relationship between III-Dv, III-D1, and III-E system systems on p6:

“It has been hypothesized that the type III-Dv complex represents an evolutionary intermediate between the multi-subunit type III-D1 system and the recently described single polypeptide type III-E system (Extended Data Fig. 1). Our structure supports this notion ...”

There are no doubts that these systems are evolutionarily related but, as mentioned by reviewer 1, type III-Dv can be either an intermediate or has undergone a faster evolution. The paragraph on p6. does not provide any argument in favor of one or another scenario. This sentence should be rephrased or other arguments given that help to discriminate between an evolutionary intermediate and a common origin but independent evolution.

2. For the modeled Mg²⁺ ions a brief description of the coordination should be provided and compared to the expected Mg²⁺ coordination number, geometry, and the distances to co-originating groups. This is a common analysis to confirm the assignment of an ion to a density. Later, RMSD values between the average positions of Mg²⁺ simulated by MD and refined within cryo-EM maps are useful to calculate and show their overlaps (or not) in a figure.

In the present manuscript, the comparison is limited by an ambiguous phrase in p9/10: “We note that the placement of the diffused Mg²⁺ ions is consistent with the experimental cryo-EM map (Fig. 3). Indeed, the ions engage in coordinating the RNA backbone and the protein residues where weak density is experimentally observed.” Which weak density do authors refer to?

3. A technical question. Self-target pre-cleavage complex and post-cleavage complex are obtained from the same grid(s). I could not find any information in Methods about the timing during the preparation of the grids. How long were the grids incubated with Mg²⁺ before plunge freezing? Is the observed pre- and post-cleaved protein ratio consistent with kinetics measured by EMSA?

4. ED Fig4. The panels g-j should be redrawn, aligned and cutoff resolution should be indicated.

Reviewer #3 (Remarks to the Author):

RNA targeting and cleavage by the type III-Dv CRISPR effector complex

This paper describes structural analysis of the type III-Dv CRISPR effector complex. This architecture of this complex is interesting as a potential evolutionary intermediate in type III effectors. A strength of the paper is the high-resolution of the structural data corresponding to different states. The following points should be addressed:

Line 162: there is no Extended Data Fig 6e

Line 179, 180. The authors state that type III-Dv complex utilizes a mechanism for target seeding distinct from previously characterized type III complexes. However, on lines 155 and 156, the authors state that the seeding mechanism likely shares similarities to that of type III-B complex.

Line 199: the term rapid is a relative term and is not meaningful without comparison; it depends on conditions, nuclease concentration, divalent cation concentration, pH, etc.

Line 210: the reaction under consideration is phosphotransesterification (or as referred to on line 252, 2'-O-transphosphorylation), not “hydrolysis” . In some cases the authors label it correctly but in other cases they refer to it as hydrolysis. These should be corrected.

Line 218: here and elsewhere the authors imply that Cas10 'produces' cOA, but in discussion they seem to indicate that it Cas10 'may produce' cOA. That is, Line 339: the authors state that type III Dv 'may' be involved in cOA signaling. This should be clarified.

Line 233: Section heading is too strong. It should be changed to: "Computational support for coordination of divalent cations by Type III-Dv active site aspartates"

Line 254: revise to 'positioning of the 2'-OH nucleophile in line with the scissile 5'-O-P bond (π catalysis),"

Line 284: Mg1B does not seem to be coordinated to O5'; it looks like it is Mg1A that is coordinating to O5'. This is an example of Lewis acid catalysis by a divalent metal ion.

Line 286 and 287: Although there are examples of nucleobases acting a general bases to activate 2'-OH as a nucleophile, I'm not aware of any examples, in which the nucleobase attached to the ribose bearing the nucleophilic 2'-OH acts in such a manner. This would be highly unusual. If there is precedence for this, the authors should cite the primary literature. Same point for line 301.

Lines 290 and 291: it is also possible that the positive charge from the metal ioin and the R residues could lower the pKa of the 2'-OH electrostatically.

Line 296: Indicate what atom the actual structural data show Mg3A to be close to, then speculate about what it's role could be.

Line 308: In this section, when the authors indicate that the active sites of the type III RNases resemble ribozymes, they appear to be talking mostly about the in-line geometry. This geometry is necessary for the actual chemistry, so it s not clear that there is much substance to the point that type III RNases resemble ribozymes.

Lines 315 and 316: when the pdb codes are given, is a literature reference supposed to be given as well?

Line 356, 362: I suggest that the authors replace “analysis” with consideration. It is important to remember that structural observations provide a framework to test hypothesis about mechanism including acid and base catalysis but structure alone does not establish the mechanism. The authors cannot assess whether the active sites are reflecting a pre-catalytic state that undergoes local changes along the reaction path. While it is reasonable and appropriate to make hypotheses about the mechanism based on the observed structure, these statements should always be viewed as hypotheses, unless functional work can establish a deeper connection. For example, a loss of activity upon mutating an Arg to Ala would be consistent with some important role in catalysis, but it would be necessary to design tests that link the Arg to pH dependence or to one of the atoms along the reaction coordinate to establish it in general acid-base catalysis.

Reviewer #4 (Remarks to the Author):

The authors report structures of type III-Dv complex in 4 different states that show metal ion coordination in the active sites. The authors then use structural, biochemical and computational analysis to interpret mechanism and make comparisons to more established mechanisms of metal-containing small self-cleaving ribozymes. While the experimental data is valuable, in many cases the mechanistic interpretation is not fully supported and the discussion in the context of known ribozyme mechanisms has many important omissions. If the major concerns below could be addressed in a revised manuscript I would feel the work would be appropriate for Nat. Commun.

The major points of concern are summarized:

1. The authors indicate in the abstract and the main text that the catalytic residues of the type II complexes resemble the active site of ribozymes including the hammerhead, pistol and Varkud satellite ribozymes. What exactly does this mean? Do the active sites exhibit the characteristic common L-platform/L-scaffold architecture of these ribozymes which are all of the “G+M” class? This is poorly discussed in the paper and not appropriately referenced – the identification of the common architecture was reported in RNA (2020) 26, 111-126.
2. The authors discuss RNA cleavage and reference Breaker’s “speed limits” paper – reference 57. Later, Break wrote a perspective in ACS Chemical Biology (ACS Chem. Biol. 2017, 12, 4, 886–891) that called for the community studying RNA-cleaving enzymes to consider adopting common terms and conventions and “clean up debris” associated with the literature in this area. The community responded to Breaker’s call with a report of an ontology for facilitating discussion of catalytic strategies of RNA-cleaving enzymes – reference 60 in the paper – that extends the scope of Breaker’s original work and should probably be

referenced first along with reference 57 on p. 10 line 257 in the section “RNA cleavage occurs through 2'-O-transphosphorylation”.

3. From the description of the computational methods, it seems the Mg(II) ion model was that of Li and Merz (12-6-4). This model was parameterized as an isolated ion in water, and never trained with respect to its interactions with amino or nucleic acid residues. So while it gives quite good solution properties in the infinite dilution limit [J. Comput. Chem. (2015) 36, 970-982], and if pairwise corrections are not used, gives nonsense for interaction with most ionic ligands including phosphate oxygens. Such pairwise corrections to the Li-Merz 12-6-4 model were published for several divalent ions that have balanced interactions with phosphates [J. Phys. Chem. B (2015) 119, 15460-15460]. It is hard to judge the reliability of the predictions for Mg(II) for interactions with negatively charged ligands including phosphates when a model was used that is known to be quantitatively inaccurate for these interactions.

4. The authors claim on p. 10, line 247 with regard to a Mg(II) predicted from simulations to be coordinating the scissile phosphate and an aspartate residue: “Abrogation of RNA cleavage through mutagenesis of these conserved aspartate residues indicates that this coordination is essential for RNA cleavage in all 3 sites of the type II-Dv complex, revealing a key, conserved role for catalytic aspartates across all type III systems.” If the role is meant to imply coordination of a Mg(II) ion, this seems somewhat speculative. Would it not be better to chemically test the non-bridge phosphoryl oxygen with a phosphorothioate to test for a normal thio effect that can be rescued by more thiophilic metals such as Cd(II). This was how a key structural Mg(II) ion, also not apparent from X-ray density but predicted from 3D-RISM as well as molecular simulations, was identified in the active site of the Varkud satellite ribozyme active site [Nat. Chem. (2020) 12, 193-201].

5. The discussion on p. 11 of “Acid-base catalysis differs between the three active sites” spends a lot of time speculating without very convincing evidence of possible acid-base mechanisms. If the authors propose that Mg(II) is acting as an acid, one should examine activity-pH profiles for similarly-sized divalent metal ions that have diverse pKa values, as was done for the pistol ribozyme to demonstrate a small self-cleaving ribozyme employed a metal-bound water molecule as the acid [JACS (2019) 141, 7865–7875]. The hammerhead ribozyme, like pistol and VS ribozymes, also contains a Mg(II) coordinating the scissile phosphate in the active state (although in the crystal structure it occupies a different binding mode). The metal acts to facilitate acid catalysis by activating another 2'OH (G8, not the nucleophile) to then act as the acid as predicted by computation [JACS (2008) 130, 3053-3064] and confirmed experimentally [JACS (2009) 131, 1135–1143]. It may be possible that the Mg(II) ions play a similar role in the acid step here by pKa shifting other residues. It is of interest that the HDV ribozyme also binds a Mg(II) ion at the scissile phosphate in the active site, and like the hammerhead uses the Mg(II) to pKa shift a 2'OH – but in this case the nucleophile, which then is able to be deprotonated in a pre-equilibrium step. This results in a dissociative-like mechanism confirmed by kinetic isotope effect measurements [JACS (2023) 145, 2830-2839]. Could the Mg(II) in site 1 or 3 be playing a similar role to assist deprotonation of the nucleophile?

6. What is the experimental evidence that would support a metal-bound hydroxide would serve as a general base, and this would cause an unfavorable pKa shift relative to an solvated hydroxide ion, and this distinction should be apparent from the activity-pH profile.

7. The discussion on p. 11-12 “Type III RNase active sites resemble ribozymes” is unclear and not supported or referenced. The figure (Fig. 4) shown does not convey this message very well. The X-ray structures for the hammerhead, pistol and VS ribozymes do not show the Mg(II) binding sites and in any event do not represent the catalytically active states of the ribozymes, so it is unclear what is being compared. The authors should consult ACS Catal. (2018) 8, 314-327 for a comparative analysis of these ribozymes active sites, and also reference the original papers corresponding to the ribozyme PDB structures in Fig. 4.

1 REVIEWER COMMENTS

3 Reviewer #1 (Remarks to the Author):

The manuscript by Schwartz et al. determined a series of CryoEM structures (+/- bound target
RNA and pre- and post- target RNA cleavage) of a newly identified and biologically rare, type III-
Dv complex from Synechocystis. Previously, the Koonin lab had identified this III-D complex and
proposed to be a to be a possible evolutionary intermediate between multi-subunit type III
crRNP complexes and the recently characterized single-subunit III-E complex (but it may simply
be a highly divergent and recently evolved type III system). The complex has multiple domain
fusions like the type III-E complex but unlike type III-E, has Cas10, Cas5 and Csx19 subunits
and lacks TPR-CHAT protease and other III-E-specific subunits. As a major focus of the study,
the authors focused on determining the mechanism of type III target RNA cleavage using a
combination of cryo-EM, molecular dynamics simulations, and biochemical assays. More
specifically, they sought to understand how each of the three catalytic sites responsible for
target RNA cleavage carry phosphodiester bond cleavage reactions and modeled predicted
catalytic Mg ions examined turnover of the cleaved target RNA.

Comments:

1) No information is presented on the in vivo function of the complex (even in a heterologous E.
coli host if the native organism is not genetically amenable) to excite us about the observed
structural features presented. Importantly, this complex has a Cas10 subunit which in other
Type III systems is capable of ssDNA cleavage (via HD domain) and/or generating cyclic
oligoadenylate signaling molecules using ATP (via Palm domain). The authors now mention that
a catalytically active PALM active site (GGDD) is predicted in the Cas10 but there is still no
description on if the Cas10 is predicted to be catalytically active or inactive for for
ssDNase activity based on amino acid sequence features (e.g. hallmark HD motif, etc.; this
should be clearly noted). There is no description or experimental investigation on Cas10 from
this complex.

While we agree that a detailed structural and mechanistic understanding of cOA signalling by
this complex is outside of the scope of this manuscript, the manuscript would be significantly
strengthened if the authors experimentally demonstrated/addressed that this complex is capable
of cOA signalling and/or ssDNase activity. Experimentally evaluating/demonstrating these key
predicted Cas10 activities is key to our fuller understanding of the proposed evolutionary link
between this system and other type III systems. Moreover, this would provide foundational
knowledge toward future work aimed ultimately at gaining detailed insight into these important
activities that likely provide immunity against the viruses (The Cas7-directed RNase activity
alone is likely insufficient for immunity and clearance of a DNA virus or plasmid).

We appreciate the concern of the reviewer. To address this, we have tested the ability of the III-
Dv effector complex to produce cOA and activate the ancillary nuclease Csx1. Activated Csx1
subsequently cleaves a fluorophore-quencher RNA, resulting in an increased fluorescence
signal. As expected, we observe cOA production in the presence of a target RNA, but not in the
presence of non-cognate RNA. Furthermore, mutation of the Cas10 Palm domain active site
abrogates this activity. This is now included in the manuscript as Extended Data Fig 6e, and
referenced in the results section as follows:

"We tested the ability of Cas10 within the III-Dv effector complex to produce cOA and activate
the ancillary nuclease Csx1. As expected, target RNA but not non-cognate RNA activates
Cas10 to produce cOA which in turn activates the non-specific RNase activity of Csx1, and

mutation of the Cas10 Palm domain catalytic residues abrogates this (Extended Data Fig. 6e).
Therefore, we propose the differences in Cas10 structure with self and non-self RNA targets
bound correspond to different activation states.”

We have also slightly modified the discussion to reflect that Cas10 conformational changes are
consistent with Palm domain activation and the Palm domain is involved in producing cOAs.

2) It seems illogical to be describing conformational changes in Cas10 required for “activation”
(extended figure 6) if there are no activities for this protein being examined (or is the
conformational change of Cas10 in this system essential for target RNA cleavage? If so, the
evidence should be explicitly described).

Since we have now added this data, we feel that it is appropriate to use this description.

3) Line 205: reference #8 should be included in addition to refs 6 and 7. Line 343: Likewise, ref
8 should be included.

These references have been added.

**Reviewer #2** (Remarks to the Author):

The revised manuscript presents an improved version of the initial submission. A few points
listed below still need clarification or improvement before publishing.

1. The paragraph describing the evolutionary relationship between III-Dv, III-D1, and III-E
system systems on p6:

“It has been hypothesized that the type III-Dv complex represents an evolutionary intermediate
between the multi-subunit type III-D1 system and the recently described single polypeptide type
III-E system (Extended Data Fig. 1). Our structure supports this notion ...”

There are no doubts that these systems are evolutionarily related but, as mentioned by reviewer
1, type III-Dv can be either an intermediate or has undergone a faster evolution. The paragraph
on p6. does not provide any argument in favor of one or another scenario. This sentence should
be rephrased or other arguments given that help to discriminate between an evolutionary
intermediate and a common origin but independent evolution.

We have amended the text as follows:

“Our structure supports this evolutionary relationship, as the organization of the type III-D1
operon is maintained and subunits are physically fused together through flexible ~20-residue
linker polypeptides.”

2. For the modeled Mg²⁺ ions a brief description of the coordination should be provided and
compared to the expected Mg²⁺ coordination number, geometry, and the distances to co-
originating groups. This is a common analysis to confirm the assignment of an ion to a density.
Later, RMSD values between the average positions of Mg²⁺ simulated by MD and refined
within cryo-EM maps are useful to calculate and show their overlaps (or not) in a figure.

We thank the reviewer for raising this important point, which was not completely clear in our
previous version of the manuscript. In this study, we used molecular dynamics (MD) simulations
to illustrate the diffusion from bulk solvent to the active site, as shown in Extended Data Fig 8a,

b. Subsequent refinement was achieved through a mixed quantum-classical (QM/MM) approach,
which enabled to describe the coordination of the metal ions in each active site at a Density
Functional theory (DFT) level. As illustrated in Extended Data Fig 8, Mg^{2+} ions display a typical
octahedral coordination sphere. Extended Data Fig 8d reports the average distance between
each Mg^{2+} ion and its ligands. The modeled Mg^{2+} ions (shown as green spheres in Fig 3d) were
then aligned with the non-proteinaceous density (gray, Fig 3d). This alignment underscores the
agreement with the non-proteinaceous density and the importance of conserved aspartate
residues in the coordination of Mg^{2+} ions. We have now corrected the figure number citation in
the revised manuscript.

In the present manuscript, the comparison is limited by an ambiguous phrase in p9/10: “We note
that the placement of the diffused Mg^{2+} ions is consistent with the experimental cryo-EM map
(Fig. 3). Indeed, the ions engage in coordinating the RNA backbone and the protein residues
where weak density is experimentally observed.” Which weak density do authors refer to?

In the presence of Mg^{2+} ions, we observed non-proteinaceous density around conserved Asp
residues (shown in gray color, Fig 3d), while this density was not observed in the absence of
119 Mg^{2+} . While this suggests coordination with metal ions, we couldn't definitively assign the
120 observed density to Mg^{2+} based on cryo-EM maps alone. To ascertain the location of Mg^{2+} ions,
we carried out extensive computational studies, including mixed quantum-classical approaches
(Extended Data Fig 8), which confirmed the location of Mg^{2+} ions in line with the non-
proteinaceous density, and in coordination with the surrounding Asp residues. We have now
revised the legend of Fig. 3 in the revised manuscript as suggested.

3. A technical question. Self-target pre-cleavage complex and post-cleavage complex are
obtained from the same grid(s). I could not find any information in Methods about the timing
during the preparation of the grids. How long were the grids incubated with Mg^{2+} before plunge
freezing? Is the observed pre- and post-cleaved protein ratio consistent with kinetics measured
by EMSA?

The grid was prepared after 16.5 min incubation with Mg^{2+} . At this time, roughly half of the
target has been cleaved (based on Ext Data Fig 7D). This does not correspond to the ratio of
particles in the final pre- and post-cleavage reconstructions (~40,000 and ~180,000,
respectively). This is not unusual in cryo-EM, since it is often necessary to discard large
amounts of particles throughout the processing pipeline to find a homogeneous subset that can
be refined to high resolution. It is likely that the post-cleavage state is more heterogeneous than
the pre-cleavage state, so to identify a sufficiently homogenous population for refinement more
particles needed to be discarded.

The methods section now specifies the length of incubation with Mg^{2+} prior to vitrification.

4. ED Fig4. The panels g-j should be redrawn, aligned and cutoff resolution should be indicated.

This has been replaced.

**Reviewer #3** (Remarks to the Author):

RNA targeting and cleavage by the type III-Dv CRISPR effector complex

This paper describes structural analysis of the type III-Dv CRISPR effector complex. This
architecture of this complex is interesting as a potential evolutionary intermediate in type III

effectors. A strength of the paper is the high-resolution of the structural data corresponding to
different states. The following points should be addressed:

Line 162: there is no Extended Data Fig 6e

This has been corrected to Extended Data Fig 5d,e.

Line 179, 180. The authors state that type III-Dv complex utilizes a mechanism for target
seeding distinct from previously characterized type III complexes. However, on lines 155 and
156, the authors state that the seeding mechanism likely shares similarities to that of type III-B
complex.

We apologize for the confusion. We are making the point that while there are similarities in
target seeding between type III effector complexes (namely the 3' seed), based on our structural
and biochemical data, the III-Dv effector complex uses a distinct mechanism, where the 3' end
is held in a conformation amenable to efficient hybridization with target RNA. We have amended
the text to improve clarity as follows:

"In summary, while type III effector complexes may use similar overall strategies for seeding
target RNA binding (namely the 3' location of the seed region), based on our structural and
biochemical data the type III-Dv effector complex uses a distinct mechanisms whereby the 3'
end is of the crRNA is held in a conformation amenable for efficient target hybridization."

Line 199: the term rapid is a relative term and is not meaningful without comparison; it depends
on conditions, nuclease concentration, divalent cation concentration, pH, etc.

We have replaced this with the term "efficient".

Line 210: the reaction under consideration is phosphotransesterification (or as referred to on
line 252, 2'-O-transphosphorylation), not "hydrolysis" . In some cases the authors label it
correctly but in other cases they refer to it as hydrolysis. These should be corrected.

This is indeed an important point. We have now substituted the word "hydrolysis" with
"cleavage."

Line 218: here and elsewhere the authors imply that Cas10 'produces' cOA, but in discussion
they seem to indicate that it Cas10 'may produce' cOA. That is, Line 339: the authors state that
type III Dv 'may' be involved in cOA signaling. This should be clarified.

We now provide experimental data to validate this.

Line 233: Section heading is too strong. It should be changed to: "Computational support for
coordination of divalent cations by Type III-Dv active site aspartates"

We have changed the heading.

Line 254: revise to 'positioning of the 2'-OH nucleophile in line with the scissile 5'-O-P bond (α
catalysis),"

We have included the suggested changes.

Line 284: Mg1B does not seem to be coordinated to O5'; it looks like it is Mg1A that is
coordinating to O5'. This is an example of Lewis acid catalysis by a divalent metal ion.

We thank the reviewer for pointing out these mistakes, which we have corrected.

Line 286 and 287: Although there are examples of nucleobases acting a general bases to
activate 2'-OH as a nucleophile, I'm not aware of any examples, in which the nucleobase
attached to the ribose bearing the nucleophilic 2'-OH acts in such a manner. This would be
highly unusual. If there is precedence for this, the authors should cite the primary literature.
Same point for line 301.

The reviewer is correct. To the best of our knowledge, there is no precedence for the
nucleobase attached to the ribose bearing the nucleophilic 2'-OH, acting as a base. Hence, we
removed this claim in the revised manuscript. We considered this possibility based on
Bevilacqua and coworkers (*ACS Catal.* **2018**, *8*, 314–327) where they considered atoms within
5 Å of the 2'-OH as contacting atoms and discussed the role of all such atoms in the catalysis.

The role of hydrated metal ions acting as acids is reported for the pistol ribozyme (*J. Am. Chem.*
*Soc.* **2019**, *141*, 7865–7875)

Lines 290 and 291: it is also possible that the positive charge from the metal ion and the R
residues could lower the pKa of the 2'-OH electrostatically.

This is a possibility. Within the active site (Extended Data Fig. 10) a possible hydrogen bond
donation to the 2'-OH could indeed lower its pKa (*ACS Catal.* **2018**, *8*, 314–327). We have also
revised the mechanism accordingly.

Line 296: Indicate what atom the actual structural data show Mg3A to be close to, then
speculate about what it's role could be.

We have included the role of Mg3A in the revised manuscript.

Line 308: In this section, when the authors indicate that the active sites of the type III RNases
resemble ribozymes, they appear to be talking mostly about the in-line geometry. This geometry
is necessary for the actual chemistry, so it s not clear that there is much substance to the point
that type III RNases resemble ribozymes.

We thank the reviewer for correctly pointing out this shortcoming in the manuscript. We have
added the comparison between type III-Dv and ribozymes.

Lines 315 and 316: when the pdb codes are given, is a literature reference supposed to be
given as well?

This is not needed. The PDB is the reference.

Line 356, 362: I suggest that the authors replace “analysis” with consideration. It is important to
remember that structural observations provide a framework to test hypothesis about mechanism
including acid and base catalysis but structure alone does not establish the mechanism. The
authors cannot assess whether the active sites are reflecting a pre-catalytic state that

undergoes local changes along the reaction path. While it is reasonable and appropriate to
make hypotheses about the mechanism based on the observed structure, these statements
should always be viewed as hypotheses, unless functional work can establish a deeper
connection. For example, a loss of activity upon mutating an Arg to Ala would be consistent with
some important role in catalysis, but it would be necessary to design tests that link the Arg to pH
dependence or to one of the atoms along the reaction coordinate to establish it in general acid-
base catalysis.

We have used word consideration in the revised manuscript.

**Reviewer #4** (Remarks to the Author):

The authors report structures of type III-Dv complex in 4 different states that show metal ion
coordination in the active sites. The authors then use structural, biochemical and computational
analysis to interpret mechanism and make comparisons to more established mechanisms of
metal-containing small self-cleaving ribozymes. While the experimental data is valuable, in
many cases the mechanistic interpretation is not fully supported and the discussion in the
context of known ribozyme mechanisms has many important omissions. If the major concerns
below could be addressed in a revised manuscript I would feel the work would be appropriate
for Nat. Commun.

The major points of concern are summarized:

1. The authors indicate in the abstract and the main text that the catalytic residues of the type II
complexes resemble the active site of ribozymes including the hammerhead, pistol and Varkud
satellite ribozymes. What exactly does this mean? Do the active sites exhibit the characteristic
common L-platform/L-scaffold architecture of these ribozymes which are all of the “G+M” class?
This is poorly discussed in the paper and not appropriately referenced – the identification of the
common architecture was reported in RNA (2020) 26, 111-126.

This is indeed a crucial aspect. We have addressed this point by including the discussion of the
active sites of the ribozymes and type III-Dv and made the necessary revisions to the references
as suggested.

2. The authors discuss RNA cleavage and reference Breaker’s “speed limits” paper – reference
57. Later, Break wrote a perspective in ACS Chemical Biology (ACS Chem. Biol. 2017, 12, 4,
886–891) that called for the community studying RNA-cleaving enzymes to consider adopting
common terms and conventions and “clean up debris” associated with the literature in this area.
The community responded to Breaker’s call with a report of an ontology for facilitating discussion
of catalytic strategies of RNA-cleaving enzymes – reference 60 in the paper – that extends the
scope of Breaker’s original work and should probably be referenced first along with reference 57
on p. 10 line 257 in the section “RNA cleavage occurs through 2’-O-transphosphorylation”.

We have included both references as suggested.

3. From the description of the computational methods, it seems the Mg(II) ion model was that of
Li and Merz (12-6-4). This model was parameterized as an isolated ion in water, and never trained
with respect to it’s interactions with amino or nucleic acid residues. So while it gives quite good
solution properties in the infinite dilution limit [J. Comput. Chem. (2015) 36, 970-982], and if
pairwise corrections are not used, gives nonsense for interaction with most ionic ligands including

phosphate oxygens. Such pairwise corrections to the Li-Merz 12-6-4 model were published for
several divalent ions that have balanced interactions with phosphates [J. Phys. Chem. B (2015)
119, 15460-15460]. It is hard to judge the reliability of the predictions for Mg(II) for interactions
with negatively charged ligands including phosphates when a model was used that is known to
be quantitatively inaccurate for these interactions.

We agree with the reviewer's assessment that the Li and Merz (12-6-4) parameters exhibit
superior performance compared to other parameters, particularly in terms of diffusion and
kinetics, as highlighted in: (i) *J. Comput. Chem.* **2015**, 36, 970-982 and (ii) *J. Chem. Theory*
*Comput.* **2020**, 16, 3, 1913-1923. Notably, the latter study demonstrates the efficacy of the Li and
Merz 12-6-4 parameters for divalent ion protein interactions, surpassing other force fields. These
parameters underwent further refinement to enhance their applicability to nucleic acids (*J. Phys.*
*Chem. B* **2015**, 119, 15460-15460) and proteins (*J. Chem. Theory Comput.* **2022**, 18, 2367-2374).
However, it's crucial to note that the aforementioned parameters were originally developed and
tested independently for either amino acids or nucleic acids, not both.

In this work, Mg²⁺ ions of the bulk solvent were described through the Li and Merz 12-6-4
parameters, obtaining the diffusion of these ions into each catalytic site. Subsequently, QM/MM
simulations were applied to describe the coordination of these ions in each of the active sites
using a DFT level of theory (Extended Data Fig. 9).

4. The authors claim on p. 10, line 247 with regard to a Mg(II) predicted from simulations to be
coordinating the scissile phosphate and an aspartate residue: "Abrogation of RNA cleavage
through mutagenesis of these conserved aspartate residues indicates that this coordination is
essential for RNA cleavage in all 3 sites of the type II-Dv complex, revealing a key, conserved
role for catalytic aspartates across all type III systems." If the role is meant to imply coordination
of a Mg(II) ion, this seems somewhat speculative. Would it not be better to chemically the the
non-bridge phosphoryl oxygen with a phosphorothioate to test for a normal thio effect that can
be rescued by more thiophilic metals such as Cd(II). This was how a key structural Mg(II) ion,
also not apparent from X-ray density but predicted from 3D-RISM as well as molecular
simulations, was identified in the active site of the Varkud satellite ribozyme active site [Nat.
Chem. (2020) 12, 193-201].

We thank the reviewer for pointing out this interesting study (Nat. Chem. **2020**, 12, 193-201, now
cited). The study of thio effects in transphosphorylation proved to be highly informative in clarifying
metal binding, particularly in the case of the Varkud Satellite (VSr) ribozyme where the
crystallographic density for Mg²⁺ in the active site was not apparent. Molecular dynamics
simulations provided additional confirmation of the crucial role played by Mg²⁺ in organizing the
active site, favorable for α and γ catalysis.

In our study, we observed non-proteinaceous density around conserved Asp residues when in
the presence of Mg²⁺. This non-proteinaceous density was absent in the Mg²⁺-free map. While
this suggests the presence of Mg²⁺ coordinated by Asp residues, the weak nature of the density
hampered the unambiguous location of the ions. To ascertain Mg²⁺ placement, we employed
molecular dynamics (MD) simulations, revealing the diffusion of Mg²⁺ ions from the bulk solvent
to the active sites. Further refinement was obtained through mixed quantum-classical (QM/MM)
simulations, describing the active sites at a DFT level, which confirmed the location of Mg²⁺ ions
in line with the non-proteinaceous density, and in coordination with the surrounding Asp residues
(Fig. 3d, Extended Data Fig. 8). This affirms the significance of conserved aspartate residues in
the coordination of Mg²⁺, a commonality in several Cas proteins and in other metalloproteins. We

also recall that catalysis is hampered in the absence of divalent ions, which affirms their catalytic
role. Also, a negligible structural difference at the level of the active sites (RMSD < 1 Å between
structures with and without Mg²⁺) suggests that the metal ions hold a marginal structural role, at
odds with the VS ribozyme.

Finally, we agree with the reviewer that metal rescue experiments would provide valuable insights
into the catalytic processes. Here, we primarily focused on elucidating the structural arrangement
of the type III-Dv CRISPR system and its active sites, which also enabled us to formulate possible
hypotheses on catalysis. We intend to delve deeper into the catalytic mechanism in subsequent
studies, which will require more in-depth investigations. Due to the complexity of these studies,
we believe they are beyond the scope of the current work.

5. The discussion on p. 11 of “Acid-base catalysis differs between the three active sites” spends
a lot of time speculating without very convincing evidence of possible acid-base mechanisms. If
the authors propose that Mg(II) is acting as an acid, one should examine activity-pH profiles for
similarly-sized divalent metal ions that have diverse pK_a values, as was done for the pistol
ribozyme to demonstrate a small self-cleaving ribozyme employed a metal-bound water molecule
as the acid [JACS (2019) 141, 7865–7875]. The hammerhead ribozyme, like pistol and VS
ribozymes, also contains a Mg(II) coordinating the scissile phosphate in the active state (although
in the crystal structure it occupies a different binding mode). The metal acts to facilitate acid
catalysis by activating another 2'OH (G8, not the nucleophile) to then act as the acid as predicted
by computation [JACS (2008) 130, 3053-3064] and confirmed experimentally [JACS (2009) 131,
1135–1143]. It may be possible that the Mg(II) ions play a similar role in the acid step here by pK_a
shifting other residues. It is of interest that the HDV ribozyme also binds a Mg(II) ion at the scissile
phosphate in the active site, and like the hammerhead uses the Mg(II) to pK_a shift a 2'OH – but
in this case the nucleophile, which then is able to be deprotonated in a pre-equilibrium step. This
results in a dissociative-like mechanism confirmed by kinetic isotope effect measurements [JACS
(2023) 145, 2830-2839]. Could the Mg(II) in site 1 or 3 be playing a similar role to assist
deprotonation of the nucleophile?

We thank the reviewer for this very informative analysis. While we conducted RNA cleavage
experiments across different pH levels (Extended Data Fig. 10b), this experiment alone is
insufficient to establish the metal's role as an acid or base. We have thereby introduced that “we
examined the active sites in our structure to hypothesize the reaction mechanism”, and scaled
back on speculative claims in the “Acid-base catalysis differs between the three active sites”
section. We agree with the reviewer that activity-pH profiles for similarly-sized divalent metal ions
with varying pK_a values would provide crucial insights into the role of metal ions in catalysis. In
the current study, our primary focus was on unraveling the structural arrangement and developing
catalytic hypotheses based on this structural understanding and further computational
investigations. Hence, we revised the heading into “Structural basis of acid-base catalysis and
ribozyme resemblance in Type III-Dv active sites”. Our future endeavors will be directed towards
in-depth investigations of the catalytic mechanism through dedicated experimental studies and
extensive QM/MM and free-energy simulations. This will explore the catalytic possibilities and
establish the cleavage mechanisms in the three active sites.

Our RNA cleavage experiments across different pH levels reported a diminished catalytic activity
with higher pK_a Ca²⁺ ions, and an enhancement of the catalysis with lower pK_a Ni²⁺ ions,
suggesting that Mg²⁺ ions could potentially act as acid in the catalysis at site 1 and site 2
(Extended Data Fig. 10b). An in-depth analysis of each active site for the γ' and γ” catalysis was
also carried out, drawing inspiration from Bevilacqua and coworkers (ACS Catal. 2018, 8,

314–327). These refinements and additions aim to provide a more in-depth discussion based on
our observations.

6. What is the experimental evidence that would support a metal-bound hydroxide would serve
as a general base, and this would cause an unfavorable pKa shift relative to an solvated
hydroxide ion, and this distinction should be apparent from the activity-pH profile.

We agree with the reviewer's comment, and we removed this claim from the manuscript.

7. The discussion on p. 11-12 “Type III RNase active sites resemble ribozymes” is unclear and
not supported or referenced. The figure (Fig. 4) shown does not convey this message very well.
The X-ray structures for the hammerhead, pistol and VS ribozymes do not show the Mg(II)
binding sites and in any event do not represent the catalytically active states of the ribozymes,
so it is unclear what is being compared. The authors should consult ACS Catal. (2018) 8, 314-
327 for a comparative analysis of these ribozymes active sites, and also reference the original
papers corresponding to the ribozyme PDB structures in Fig. 4.

We have addressed this point by incorporating a discussion in the manuscript, comparing
ribozymes and the type III-Dv active site. Additionally, we have updated Fig. 4b accordingly.

REVIEWERS' COMMENTS

Reviewer #1 (Remarks to the Author):

Demonstration that the complex is capable of generating cyclic adenylate signalling molecules that activate the csx1 subunit RNase activity in a target RNA-dependent manner has strengthened the paper. These findings provide important insight into the evolutionary relationships of this system relative to other type III (and perhaps type I) systems as well as provides a foundation for future work aimed at understanding how the complex functions and is regulated in vivo for basic and applied research.

For unclear reasons, the authors did not experimentally address if the target RNA-bound complex also cuts ssDNA as is the case for several other Cas10-containing type III complexes. Minimally, the authors need to make it explicitly clear if the Cas10 subunit is predicted to contain an active or inactive HD ssDNase active site based on comparisons with Cas10 structures and sequences that have been shown to be active DNases. Whatever conclusion is derived about the potential of their studied system to cleave ssDNA should be related to whether or not this is a predicted general property of known type III-D subtypes.

Reviewer #2 (Remarks to the Author):

in the revised manuscript, the authors answered all my questions and requests. I have no further comments. The manuscript can be published now.

Reviewer #4 (Remarks to the Author):

The authors have overall done a reasonable job at addressing the concerns of the previous review. One area that still needs consideration is in the clarification of the 12-6-4 models for ions. Here is the issue: the "off-the-shelf" models of Li and Merz were parameterized in bulk water at infinite dilution – there was no consideration whatsoever for interactions with other residues. Several groups have worked to correct this using pairwise corrections – these essentially over-ride the traditional "Lorentz-Berthelot" combining rules such that the A_{ij} , B_{ij} and C_{ij} terms are simply explicit pairwise parameters for the more essential atom types so that the interactions are appropriately balanced with key residues of protein or nucleic acids. The authors suggest that there is an issue using these adjusted parameters because they

were done so independently for nucleic acids [JPCB 119, 15460 (2015)] and proteins JCTC 18, 2367 (2022)] but not both at the same time. Specifically, they state in the rebuttal: “However, it's crucial to note that the aforementioned parameters were originally developed and tested independently for either amino acids or nucleic acids, not both.” They use this argument to justify the use of the 12-6-4 parameters that did not consider any corrections for either proteins or nucleic acids. This does not make sense. The pairwise corrections are at the atom-type level for which there is no overlap between proteins or nucleic acids, so they can both be used at the same time. Further, their inclusion does not alter at all the interactions with other species such as water from which the existing parameters were developed. One would think that using non-interfering pairwise parameters that made accurate interactions with water in addition to binding sites in proteins and nucleic acids independently would be better than using a set of parameters that took neither proteins or nucleic acids into consideration in any way.

To be clear – I do not think the authors need to re-run everything using a more justified set of 12-6-4 parameters (as they go on to do QM/MM), but I do think they need to point out that a better choice of pairwise 12-6-4 ion parameters exist and would have been an appropriate (and preferable) alternative to use. If this is completely ignored as it currently is, it sets a poor scientific precedent that others will read and follow, inevitably repeating mistakes from the past that Joung and Cheatham pointed out early on. So in terms of good science, this should be pointed out in the Methods section on p. 35, and mention the references where the pairwise parameters have been developed.

Response to final comments from reviewers

Reviewer 1

Demonstration that the complex is capable of generating cyclic adenylyate signalling molecules that activate the csx1 subunit RNase activity in a target RNA-dependent manner has strengthened the paper. These findings provide important insight into the evolutionary relationships of this system relative to other type III (and perhaps type I) systems as well as provides a foundation for future work aimed at understanding how the complex functions and is regulated in vivo for basic and applied research.

For unclear reasons, the authors did not experimentally address if the target RNA-bound complex also cuts ssDNA as is the case for several other Cas10-containing type III complexes. Minimally, the authors need to make it explicitly clear if the Cas10 subunit is predicted to contain an active or inactive HD ssDNase active site based on comparisons with Cas10 structures and sequences that have been shown to be active DNases. Whatever conclusion is derived about the potential of their studied system to cleave ssDNA should be related to whether or not this is a predicted general property of known type III-D subtypes.

We now include this prediction in Supplementary Fig. 6. We have also included this analysis in the main text:

Phylogenetic analyses revealed that most Cas10 subunits from type III-D systems are predicted to not have an HD domain (Makarova et al 2020; Weigand et al 2024). Indeed, a large truncation was observed in our type III-Dv Cas10 when compared to a type III-A Cas10 with known HD nuclease activity (Jia et al. 2019) (Supplementary Fig. 6). While a HDD sequence was located in type III-Dv Cas10, we predict the disrupted HD pocket would inactivate the nuclease activity of this subunit.

Reviewer 4

The authors have overall done a reasonable job at addressing the concerns of the previous review. One area that still needs consideration is in the clarification of the 12-6-4 models for ions. Here is the issue: the “off-the-shelf” models of Li and Merz were parameterized in bulk water at infinite dilution – there was no consideration whatsoever for interactions with other residues. Several groups have worked to correct this using pairwise corrections – there essentially over-ride the traditional “Lorentz-Berthelot” combining rules such that the A_{ij} , B_{ij} and C_{ij} terms are simply explicit pairwise parameters for the more essential atom types so that the interactions are appropriately balanced with key residues of protein or nucleic acids. The authors suggest that there is an issue using these adjusted parameters because they were done so independently for nucleic acids [JPCB 119, 15460 (2015)] and proteins JCTC 18, 2367 (2022)] but not both at the same time. Specifically, they state in the rebuttal: “However, it’s crucial to note that the aforementioned parameters were originally developed and tested independently for either amino acids or nucleic acids, not both.” They use this argument to justify the use of the 12-6-4 parameters that did not consider any corrections for either proteins or nucleic acids. This does not make sense. The pairwise corrections are at the atom-type level for which there is no overlap between proteins or nucleic acids, so they can both be used at the same time. Further, their inclusion does not alter at all the interactions with other species such as water from which the existing parameters were developed. One would think that using non-interfering pairwise parameters that made accurate interactions with water in addition to binding

sites in proteins and nucleic acids independently would be better than using a set of parameters that took neither proteins or nucleic acids into consideration in any way.

To be clear – I do not think the authors need to re-run everything using a more justified set of 12-6-4 parameters (as they go on to do QM/MM), but I do think they need to point out that a better choice of pairwise 12-6-4 ion parameters exist and would have been an appropriate (and preferable) alternative to use. If this is completely ignored as it currently is, it sets a poor scientific precedent that others will read and follow, inevitably repeating mistakes from the past that Joung and Cheatham pointed out early on. So in terms of good science, this should be pointed out in the Methods section on p. 35, and mention the references where the pairwise parameters have been developed.

We now include the reference suggested by the reviewer and addressed these comments in the Methods section:

While the 12-6-4 potential for divalent metals reported a good agreement in infinitely diluted systems^{79,80}, the state-of-the-art by Panteva et al. provide a better description of the metal ions interacting with RNA¹⁰².